# Conformal Prediction Beyond the Horizon: Distribution-Free Inference for Policy Evaluation

Feichen Gan, Youcun Lu, Yingying Zhang[*] and Yukun Liu

KLATASDS - MOE, School of Statistics, East China Normal University

## Abstract

Reliable uncertainty quantification is crucial for reinforcement learning (RL) in high-stakes settings. We propose a unified conformal prediction framework for infinite-horizon policy evaluation that constructs distribution-free prediction intervals for returns in both on-policy and off-policy settings. Our method integrates distributional RL with conformal calibration, addressing challenges such as unobserved returns, temporal dependencies, and distributional shifts. We propose a modular pseudo-return construction based on truncated rollouts and a time-aware calibration strategy using experience replay and weighted subsampling. These innovations mitigate model bias and restore approximate exchangeability, enabling uncertainty quantification even under policy shifts. Our theoretical analysis provides coverage guarantees that account for model misspecification and importance weight estimation. Empirical results, including experiments in synthetic and benchmark environments like Mountain Car, show that our method significantly improves coverage and reliability over standard distributional RL baselines.

## 1 Introduction

**Motivation.** As reinforcement learning (RL) are increasingly deployed in high-stakes domains, such as healthcare, robotics, and autonomous systems, robust uncertainty quantification becomes essential. While traditional policy evaluation methods focus on estimating the expected return, this is insufficient when decisions must account for risk, reliability, and rare outcomes. For example, in clinical decision-making, a treatment policy may appear beneficial on average but hide adverse effects for specific patient subgroups. Even in less safety-sensitive applications such as recommendation systems or finance, overlooking uncertainty can lead to unstable behavior and degraded user experience. Prediction intervals (PIs) for returns offer a principled way to quantify uncertainty, enabling risk-aware planning and safer deployments.

This paper focuses on constructing valid PIs for **infinite-horizon** RL settings, where the return is defined as the sum of discounted rewards. In on-policy settings, PIs help assess the variability of returns under the current policy, enabling more robust policy improvement and risk-sensitive exploration. In off-policy scenarios, where evaluating a new policy offline based on an observational dataset, PIs serve to gauge the reliability of point estimation from historical data. By constructing PIs for the return, our approach improves the transparency, reliability, and robustness of RL systems across a wide range of domains.

**Challenges.** Constructing valid PIs for returns in RL is closely tied to estimating the full return distribution, as studied in Distributional RL (DRL). In principle, conditional quantiles from this distribution can be used to form PIs. However, existing DRL-based approaches often suffer from model misspecification, leading to biased or inconsistent return distribution estimates and a lack of

---

[1][*]Corresponding author: yyzhang@fem.ecnu.edu.cn.

39th Conference on Neural Information Processing Systems (NeurIPS 2025).

formal statistical guarantees. To address this, building on the framework of conformal prediction, we propose a flexible, model-agnostic methodology for constructing PIs with asymptotic coverage guarantees. Applying conformal prediction to the infinite-horizon RL setting requires substantial methodological innovation, as it poses several fundamental challenges:

- **Unobserved Returns.** In infinite-horizon RL, the return cannot be directly observed, since in practice only finite-horizon trajectories (of length $T$) are available and future rewards beyond $T$ are unobserved. Although mitigated by discounting, the truncation error remains non-negligible in offline settings when $T$ is moderate, making it challenging to evaluate prediction errors or calibrate uncertainty.

- **Temporal Dependence.** RL data are inherently sequential, violating the exchangeability assumption required by standard conformal prediction methods.

- **Distribution Shifts.** In on-policy setting, discrepancies over time lead to complex covariate shift in the state distribution. In off-policy evaluation, discrepancies between the behavior policy and the target policy also lead to covariate shift in the state-action distribution.

**Contributions.** We propose a novel, distribution-free method that integrates conformal prediction with distributional RL to construct prediction intervals for infinite-horizon returns under both on-policy and off-policy settings. Our contributions are as follows: *(1) Pseudo-Return Construction.* We develop a modular approximation scheme for unobserved returns, combining truncated rollouts with tail sampling from learned return distributions. This design is inspired by temporal-difference learning and enables calibration despite partial observability. *(2) Calibration via Experience Replay.* To mitigate temporal dependence and approximate exchangeability, we adopt experience replay and apply random subsampling to the calibration set. This design recovers approximate exchangeability, enabling valid conformal calibration. *(3) Time-Aware Weighted Subsampling.* We address distribution shifts both over time and between policies, using a simple, weighted subsampling scheme. This enables valid calibration in off-policy settings and improves efficiency in on-policy scenarios. *(4) Theoretical Guarantees.* We establish asymptotic lower bounds on coverage using Wasserstein metrics, characterizing how model bias and density ratio estimation affect conformal validity. *(5) Empirical Validation.* We demonstrate the effectiveness of our method through empirical studies on synthetic and the Mountain Car environments.

Together, these contributions extend conformal prediction to the infinite-horizon RL setting and offer a scalable, practical framework for uncertainty-aware policy evaluation.

## 1.1 Related Work

**Risk-aware RL.** RL is a framework in which an agent interacts with an unknown environment to maximize its expected total reward. Due to the intrinsic randomness of the environment, even policies with high expected returns may occasionally yield very low rewards, which can be problematic in risk-sensitive applications such as healthcare [19] or competitive games [21]. For instance, in clinical decision-making, patient responses to treatments are stochastic, making it desirable to select actions that achieve high effectiveness while minimizing the likelihood of adverse effects. To address these concerns, risk-aware RL aims to learn policies that reduce the probability of low total rewards [16], using a variety of risk measures including entropic or exponential utility [11, 22], conditional value-at-risk [25, 6], and coherent risk measures [18].

In parallel, safe RL and constrained Markov Decision Processes (MDPs) offer an alternative approach to managing uncertainty; a comprehensive survey of safe RL is provided in [14]. Unlike risk-aware MDPs, these methods do not modify the optimality criteria; instead, risk aversion is enforced through constraints on rewards or risks [5]. While both risk-aware and safe RL approaches incorporate risk considerations into policy learning, they primarily focus on modeling risk preferences and generally do not provide formal uncertainty quantification for PIs.

**Distributional RL.** Distributional RL focuses on modeling the full return distribution rather than just its expectation. Pioneering work by [2] introduces this paradigm, followed by quantile-based approaches such as Quantile Temporal Difference (QTD) learning [7, 27], which approximates return distributions via quantile regression. These methods have led to practical advances in robotics, control, and decision-making under uncertainty [1, 4, 10, 34]. However, most DRL methods provide

pointwise quantile estimates and lack formal statistical coverage guarantees, especially under model misspecification.

By integrating conformal prediction with DRL-based distribution estimation, our framework ensures asymptotic coverage for predictive intervals, even in challenging infinite-horizon settings.

**Conformal Prediction for RL.** Conformal prediction offers distribution-free confidence intervals under exchangeable data [32]. Extending it to RL is challenging due to the inherent temporal dependencies and evolving state distributions. Recent efforts have attempted to bridge this gap. Early work such as [8] applies conformal prediction to construct trajectory-level prediction intervals in finite-horizon MDPs. Building on this idea, [12] develop a weighted conformal prediction method for off-policy evaluation, using importance sampling weights to correct for distributional shifts between behavior and target policies. However, this approach suffers from the curse of horizon, as the importance weights accumulate multiplicatively over time, resulting in high variance in long-horizon settings. In parallel, [35] introduce the COPP algorithm for contextual bandits, which approximates exchangeability via pseudo-policies and trajectory subsampling; yet, its applicability is largely limited to short-horizon problems with finite discrete action spaces. [36] further analyze how temporal correlations in Markovian data affect the coverage and width of split conformal intervals. Finally, we note a growing line of work that applies adaptive conformal prediction to online safe RL settings [29, 37], which differs fundamentally from our setting.

Despite these advances, existing methods largely focus on finite-horizon scenarios or on settings with limited state or action spaces. Prior conformal RL approaches typically handle distribution shifts between behavior and target policies using trajectory-level importance weighting, which becomes computationally inefficient as the trajectory horizon grows. In contrast, our work is the first to tackle infinite-horizon off-policy prediction in general RL settings with arbitrary state and action spaces using conformal prediction. By constructing stepwise pseudo-returns and leveraging experience replay, our method scales conformal prediction to infinite-horizon settings with standard RL data and remains effective even when only partial trajectory fragments are available.

## 2 Problem Formulation

We consider the standard RL framework [2, 17, 30], where the environment is modeled as a time-homogeneous MDP, as specified in the assumptions provided in the supplementary material. Our goal is to construct distribution-free PIs for the return of a given policy in infinite-horizon settings under both on-policy and off-policy scenarios.

**Data and Setup.** Let $\mathcal{D} = \{\zeta_i\}_{i=1}^N$ be a dataset of $N$ trajectories, each consisting of $T$ time steps. For simplicity, we assume trajectories have uniform length, but our method naturally extends to variable-length settings. Each trajectory $\zeta_i = \{(S_{it}, A_{it}, R_{it})\}_{t=0}^{T-1}$ consists of the state $S_{it}$, the action $A_{it}$ and the immediate reward $R_{it}$. These transitions are generated by a **behavior policy** $\pi_b$, such that $A_{it} \sim \pi_b(\cdot \mid S_{it})$ and evolve under a transition kernel $\mathcal{P}$ with $(R_{it}, S_{i,t+1}) \sim \mathcal{P}(\cdot \mid S_{it}, A_{it})$. In healthcare applications, each trajectory corresponds to a patient, with $S_{it}$ representing clinical features, $A_{it}$ the administered treatment, and $R_{it}$ the resulting clinical response.

**Objective.** Let $\pi$ be a **target policy** of interest. The return starting from the state $s$ is defined as $G^\pi(s) = \sum_{t=0}^\infty \gamma^t R_t$, where $R_t$ is the reward at time $t$ under policy $\pi$ and $\gamma \in (0, 1)$ is the discount factor. This return captures the long-term outcome of following policy $\pi$ from state $s$. Given a new test state $S_{\text{test}}$, we aim to construct a prediction interval for $G^\pi(S_{\text{test}})$ that achieves a user-specified coverage level $1 - \alpha$. That is, we seek a set $C(S_{\text{test}})$ such that:

$$\Pr(G^\pi(S_{\text{test}}) \in C(S_{\text{test}})) \geq 1 - \alpha.$$

In healthcare applications, $G^\pi(S_{\text{test}})$ represents the long-term treatment effect for a new patient under policy $\pi$. The prediction interval thus provides a principled range of plausible outcomes for the patient, enabling informed decision-making before the policy is actually deployed in practice. In this paper, we consider two settings:

1. **On-Policy Setting:** the target policy $\pi$ is the same as the behavior policy $\pi_b$. This setting enables evaluation using in-distribution transitions, but still faces the challenges of infinite horizon and unobserved returns.

2. **Off-Policy Setting:** the target policy $\pi$ differs from $\pi_b$. In this case, the data distribution differs from that under the target policy, and appropriate corrections for distribution shift are necessary.

**Preliminaries of DRL.** The goal of DRL is to learn the distribution of returns $G^\pi(s)$ for each state $s$. Let $\eta^\pi(s)$ denote the the probability distribution of the random return. Numerous DRL methods exist for both on-policy and off-policy settings [3]. In this paper, we adopt quantile temporal difference (QTD) learning for experiments, a prominent approach within DRL. QTD seeks to approximate the return distribution by $\eta^\pi(s) \approx \frac{1}{m} \sum_{i=1}^{m} \delta_{\theta(s,i)}$, which is an equally-weighted mixture of Dirac deltas at locations $\theta(s, i)$. The aim is to have these particles approximate the $\tau_i = (2i - 1)/(2m)$-th quantiles of $\eta^\pi(s)$ for $i = 1, \ldots, m$. Like other temporal-difference methods, QTD updates its parameters $\{(\theta(s, i))_{i=1}^m\}$ using observed transitions $(S_{it}, R_{it}, S_{i,t+1})$. In continuous and high-dimensional state spaces, function approximation offers a powerful approach for modeling $\{(\theta(s, i))_{i=1}^m\}$ and generalizing across states.

**Limitations of DRL.** A naive approach to constructing PIs would be to take the empirical quantiles of $\eta^\pi(s)$, i.e. using $[\theta(s, L), \theta(s, U)]$, where $L = \lfloor (m\alpha + 1)/2 \rfloor$ and $U = m + 1 - L$. However, such DRL-based quantile intervals, referred to as **DRL-QR**, can be unreliable in finite-sample settings and do not come with formal guarantees of asymptotic validity. For instance, [3] show that the QTD algorithm converges to a limiting distribution in finite state and action spaces; yet this limiting distribution is not guaranteed to match the true return distribution, and thus the convergence provides no assurance that QTD-based prediction intervals are asymptotically valid. In continuous state and action spaces, distributional RL methods must rely on function approximation to estimate return distributions. The theoretical guarantees of these approaches consequently depend critically on the accuracy of the modeling assumptions, rendering them susceptible to potential model misspecification. To address these limitations, we develop a conformal prediction framework that **wraps** around any return distribution estimator (such as QTD), correcting for model bias and enabling finite-sample statistical guarantees.

## 3 Conformal Policy Prediction Beyond the Horizon

We propose a novel conformal prediction (CP) framework that addresses the unique challenges of uncertainty quantification in infinite-horizon RL. Our approach combines three key innovations: (1) pseudo-returns that blend finite rollouts with learned distributional tails, (2) time-aware calibration addressing both temporal dependence and distribution shifts, and (3) replay-based weighted subsampling to restore exchangeability.

### 3.1 Overview of the Conformal Framework

Our method follows the split conformal prediction paradigm, adapted to the RL setting. Given a dataset of transition tuples $\{(S_{it}, A_{it}, R_{it}, S_{i,t+1})\}$, we partition it into a **training set** $\mathcal{D}_{tr}$, used to fit a predictive model for the return distribution, and a **calibration set** $\mathcal{D}_{cal}$, used to quantify predictive uncertainty. The overall pipeline consists of four key steps illustrated in Figure 1:

1. Train a DRL model, such as QTD learning, on $\mathcal{D}_{tr}$ to construct a return distribution estimate $\hat{\eta}^\pi(s)$ and a value function estimate $\hat{v}^\pi(s)$ under the target policy $\pi$.

2. For each calibration state, construct *pseudo-returns* by combining observed rewards with samples drawn from the estimated return distribution. The procedure for generating pseudo-returns is detailed in Section 3.2.

3. Compute nonconformity scores using the pseudo-returns in the calibration set, typically using the absolute deviation from the estimated value function: $V(s) = |\widetilde{G}^\pi(s) - \hat{v}^\pi(s)|$, where $\widetilde{G}^\pi(s)$ denotes the pseudo-return.

4. Apply conformal prediction to construct a prediction interval for a new test state $S_{\text{test}}$, using weighted subsampling to adjust for distribution shifts and experience replay to approximate exchangeability by decorrelating transitions, detailed in Section 3.3.

The nonconformity score plays a central role in quantifying uncertainty and correcting for potential estimation bias. While our framework is compatible with more sophisticated nonconformity mea-

sures, such as those used in conformalized quantile regression [26], the double-quantile score [12], and various others, we use the simple absolute-error score here for clarity and illustration.

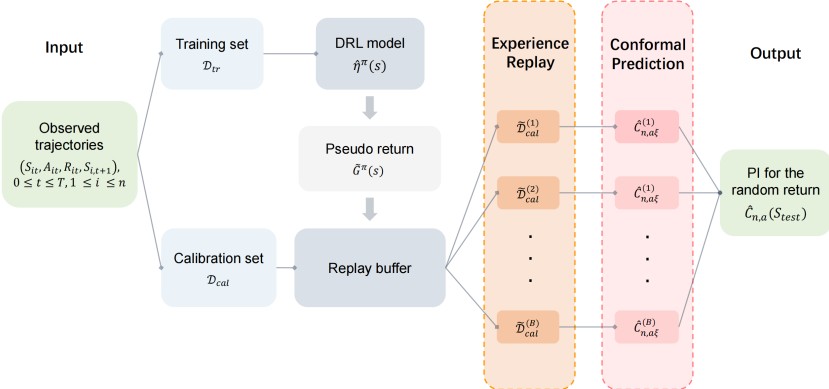

Figure 1: Pipeline of the proposed conformal policy prediction framework.

## 3.2  Pseudo-Return Construction via Truncated Rollouts

A key challenge in infinite-horizon RL is that the true return $G^\pi(s)$ is unobservable in a finite-step trajectory, making it difficult to directly evaluate nonconformity scores for conformal prediction. To address this, we introduce a novel pseudo-return construction that inspired by $k$-step temporal difference (TD) learning. We reinterpret $k$-step TD learning through the lens of distributional inference. Specifically, for each calibration point $(S_{it}, A_{it}, R_{it}, S_{i,t+1})$, we define the $k$-step *pseudo-return* as:

$$\tilde{G}^{(k)}(S_{it}) = \sum_{h=0}^{k-1} \gamma^h R_{i,t+h} + \gamma^k \tilde{G}^\pi(S_{i,t+k}), \tag{1}$$

where the first term accumulates observed rewards under the behavior policy $\pi_b$, and the second term approximates the unobserved tail using a sample from the estimated return distribution $\hat{\eta}^\pi(S_{i,t+k})$.

**Advantages.**  Pseudo-return construction approximates the infinite-horizon return using a finite rollouts combined with a bootstrapped tail. **First**, this decomposition bridges model-based and model-free RL within the conformal inference framework. **Second**, the tail value is sampled from a learned return distribution, allowing seamless integration with DRL methods such as QTD or C51 [2]. **Finally**, the rollout horizon $k$ offers a natural bias-variance trade-off: increasing $k$ incorporates more observed data, potentially reducing model bias but requiring longer rollouts; decreasing $k$ increases reliance on model predictions, offering faster calibration at the cost of higher bias.

**On-policy setting.**  We detail the QTD learning procedure for DRL used in this paper, although any DRL estimation method can be integrated into our framework. In the on-policy case, QTD estimates the return distribution conditioned on the initial state, $\hat{\eta}^\pi(s)$, via the iterative update

$$\theta(s,i) \leftarrow \theta(s,i) + \rho \cdot \frac{1}{m} \sum_{j=1}^{m} \left[ \tau_i - I(r + \gamma\theta(s',j) - \theta(s,i) < 0) \right],$$

where $\theta(s,i)$ denotes the $\tau_i$-th quantile of $\hat{\eta}^\pi(s)$, $(s,a,r,s')$ is sampled under the behavior policy $\pi$, which coincides with the target policy in the on-policy setting, and $\rho$ is a learning rate.

**Off-policy setting.**  Extending QTD to the off-policy setting requires careful modifications to account for distributional shifts between $\pi_b$ and $\pi$. We first define the return starting from a state-action pair as $G^\pi(s,a) = \sum_{t=0}^{\infty} \gamma^t R_t$, where the agent takes action $a$ in state $s$ and follows policy $\pi$ thereafter. The distribution of this return is denoted by $\eta^\pi(s,a)$. The goal of QTD is to estimate the

quantile functions of $\eta^\pi(s,a)$. The iterative update for the $\tau_i$-th quantile $\theta(s,a,i)$ is given by

$$\theta(s,a,i) \leftarrow \theta(s,a,i) + \rho \cdot \frac{1}{m} \sum_{j=1}^{m} \left[\tau_i - I(r + \gamma\theta(s',a',j) - \theta(s,a,i) < 0)\right],$$

where $\theta(s,a,i)$ is the $\tau_i$-th quantile of $\hat{\eta}^\pi(s,a)$, $(s,a,r,s')$ is sampled from the behavior policy $\pi_b$, and $a'$ is drawn from the target policy $\pi$. The result is marginalized over the action space according to $\pi$: $\hat{\eta}^\pi(s) = \sum_a \pi(a|s)\hat{\eta}^\pi(s,a)$. This modification is necessary to correct for the action distribution mismatch between behavior and target policies. For further details on distributional RL in off-policy evaluation, see [23, 15].

### 3.3 Time-Aware Calibration via Experience Replay and Weighted Subsampling

A core challenge in applying CP to RL lies in the violation of its key assumption: *exchangeability* between the calibration and test data. In RL, this is broken due to (i) temporal dependencies across transitions and (ii) distribution shifts in the state space both over time and across policies. To address these challenges, we introduce a two-pronged calibration strategy through *experience replay-based sampling* to decorrelate temporally linked transitions and *time-aware importance weighting* to correct for dynamic policy-dependent distributional shifts.

**Experience Replay.** Temporal dependence between transitions in RL makes the direct application of conformal prediction invalid. To mitigate this, we draw inspiration from deep RL techniques and treat the calibration set as a *replay buffer*, storing transition tuples $(S_{it}, A_{it}, R_{it}, S_{i,t+1})$. We then apply random subsampling from this buffer to construct approximately i.i.d. calibration samples [9]. This technique mirrors the prioritized or uniform experience replay used in deep Q-learning, effectively decorrelating transitions [28]. For the construction of $k$-step pseudo-returns, we store extended tuples of the form $\{(S_{it}, A_{it}, R_{it}, \ldots, S_{i,t+k})\}$.

**Weighted Subsampling (WS).** Instead of adopting weighted conformal prediction (WCP) [31], which is commonly used to correct for covariate shifts, we employ a *sampling-based* strategy. Specifically, we perform weighted subsampling from the calibration buffer based on estimated importance weights, producing a recalibrated set of approximately exchangeable samples tailored to the target distribution. The importance weights differ depending on whether the setting is on-policy or off-policy:

1. **On-Policy Setting.** Here, the distribution shift stems from time-indexed variation in state visitation. We define the importance weight as

$$w_{\text{on}}(s) = \frac{d\mathcal{P}_0(s)}{d\mathcal{P}_{\text{cal}}(s)} = \frac{P(\delta=1 \mid s)}{P(\delta=0 \mid s)} \frac{P(\delta=0)}{P(\delta=1)} \propto \frac{P(\delta=1 \mid s)}{P(\delta=0 \mid s)}, \tag{2}$$

   where $\mathcal{P}_0$ is the probability distribution of test states, $\mathcal{P}_{\text{cal}}$ is the marginal probability distribution over calibration states, and $\delta$ is an indicator variable, where $\delta=0$ denotes that $s$ belongs to the calibration set, and $\delta=1$ indicates that $s$ is in the test set. The second equality in Eq. (2) follows from Bayes rule, expressing the likelihood ratio as a ratio of classifier probabilities [13, 24]. In practice, $w_{\text{on}}(s)$ can be estimated using standard propensity scoring or density ratio estimation methods. In simulations, we employ logistic regression for this purpose.

2. **Off-Policy Setting.** In this case, both temporal drift and policy mismatch must be corrected. We define the importance weight over a $k$-step trajectory segment as

$$w_{\text{off}}(s_0, a_0, \ldots, s_k) \propto \frac{d\mathcal{P}_0(s_0)}{d\mathcal{P}_{\text{cal}}(s_0)} \prod_{h=0}^{k-1} \frac{\pi(a_h \mid s_h)}{\pi_b(a_h \mid s_h)}. \tag{3}$$

   This formulation adjusts for discrepancies in both state visitation and action selection between the behavior and target policies. This ratio can also be estimated using propensity scoring techniques.

To reduce the variance in PIs caused by subsampling randomness, we repeat the process $B$ times and aggregate the intervals. This technique draws from recent work in conformal prediction under distribution shift [35] and improves both coverage stability and efficiency. The complete algorithm for the on-policy setting is in Algorithm 1, while the off-policy version is deferred to the supplementary material to save space.

**Why WS Works.** In the off-policy setting, let $S_{\text{test}} := S_{\text{test},0}$ denote a test state drawn from the marginal distribution $\mathcal{P}_0(s)$, and consider the joint distribution:

$$(S_{\text{test},0}, A_{\text{test},0}, R_{\text{test},0}, \ldots, S_{\text{test},k}, G^\pi(S_{\text{test},0})) \sim \mathcal{P}_0^{\text{off}}(s_0, a_0, r_0, \ldots, s_k, G).$$

Similarly, let $\mathcal{P}_{\text{cal}}^{\text{off}}$ denote the joint distribution of rollout segments in the calibration set:

$$(S_{it}, A_{it}, R_{it}, \ldots, S_{i,t+k}, G^\pi(S_{it})) \sim \mathcal{P}_{\text{cal}}^{\text{off}}(s_0, a_0, r_0, \ldots, s_k, G).$$

The two distributions are related through the importance weight $w_{\text{off}}$, such that:

$$d\mathcal{P}_0^{\text{off}}(s_0, a_0, r_0, \ldots, s_k, G) = w_{\text{off}}(s_0, a_0, \ldots, s_k) \, d\mathcal{P}_{\text{cal}}^{\text{off}}(s_0, a_0, r_0, \ldots, s_k, G). \tag{4}$$

This identity shows that sampling from the calibration distribution according to the importance weights $w_{\text{off}}$ produces samples that approximate the test-time distribution $\mathcal{P}_0^{\text{off}}$. By reweighting the calibration set in this way, we recover approximate exchangeability between the calibration and test samples, thereby restoring the validity of conformal prediction in the presence of both temporal and policy-induced distribution shifts.

**Why Not Use WCP.** Weighted conformal prediction (WCP) typically assumes access to the full set of test-time covariates. In contrast, our setting only observes the initial state $S_{\text{test},0}$ at test time, while subsequent states $S_{\text{test},1}, S_{\text{test},2}, \ldots, S_{\text{test},k}$ remain unobserved. The WCP weight defined in Eq. 12 of [12] involves marginalizing over entire trajectories, which are unobserved. Although [12] further propose an optimization-based approximation (Eq. 14), this approach introduces additional model assumptions and tends to exhibit high variance, especially in long-horizon settings, limiting their practical applicability in our context. On the other hand, while one could adopt more elaborate designs such as that of [35] tailored for sequential decision-making, our weighted subsampling scheme offers a significantly simpler and more practical alternative, especially when only the initial states of test trajectories are observed.

## 4   Theoretical Results

In this section, we provide statistical guarantees for the PIs constructed by our method. Standard CP yields marginal coverage at level $1 - \alpha$ under the assumption of exchangeability. However, in practice, distribution shifts violate this assumption, leading to a gap between the nominal level $1 - \alpha$ and the actual coverage. Previous studies have bounded this gap using total variation distance, which fails to capture how different choices of $k$ in $k$-step rollouts affect the coverage gap. To address this, we propose a tighter upper bound on the coverage gap based on the Wasserstein distance, leveraging a recent theoretical result from [33]. Let $\mu$ and $\nu$ be two probability measures on the real space $\mathbb{R}$. For any $p > 0$, the $p$-Wasserstein distance between $\mu$ and $\nu$ is defined as $W_p(\mu, \nu) := \inf_{\kappa \in \Gamma(\mu,\nu)} \{ \int_{\mathbb{R} \times \mathbb{R}} |x - y|^p \kappa(dx, dy) \}^{1/p}$, where $\Gamma(\mu, \nu)$ denotes the set of all couplings with marginals $\mu$ and $\nu$.

Let $n$ be the cardinality of the calibration set $\mathcal{D}_{\text{cal}}$, and $\hat{\eta}^\pi(s)$ denote an estimate of the return distribution $\eta^\pi(s)$ under the target policy $\pi$. We take $\mathcal{S}$ to be the state space and define $\bar{W}_1(\eta^\pi, \hat{\eta}^\pi) := \sup_{s \in \mathcal{S}} W_1(\eta^\pi(s), \hat{\eta}^\pi(s))$. Let $\hat{w}_{\text{on}}(s)$ be an estimate of the on-policy importance weight defined in (2), and let $\widehat{C}_{N,\alpha}^{\text{on}}(\cdot)$ be the prediction interval produced by Algorithm 1. The following theorem establishes an asymptotic lower bound on the coverage in the on-policy setting.

**Condition 1.** (i) The return distribution $\eta^\pi(s)$ has a Lebesgue density bounded by $L$ for all $s \in \mathcal{S}$. (ii) $\mathbb{E}[\hat{w}_{\text{on}}(S_{it}) | \mathcal{D}_{\text{tr}}] < \infty$ and $\mathbb{E}[w_{\text{on}}(S_{it})] < \infty$ for all $0 \le t \le T - k$.

THEOREM **1** (**On-Policy Coverage Guarantee**). *Assume Condition 1, and redefine $\hat{w}_{\text{on}}(s)$ as* $\hat{w}_{\text{on}}(s) / \frac{1}{T-k+1} \sum_{t=0}^{T-k} \mathbb{E}[\hat{w}_{\text{on}}(S_{it}) | \mathcal{D}_{\text{tr}}]$ *so that* $\frac{1}{T-k+1} \sum_{t=0}^{T-k} \mathbb{E}[\hat{w}_{\text{on}}(S_{it}) | \mathcal{D}_{\text{tr}}] = 1$. *Then*

$$\lim_{n \to \infty} \Pr\left( G^\pi(S_{\text{test}}) \in \widehat{C}_{N,\alpha}^{\text{on}}(S_{\text{test}}) \right) \ge 1 - \alpha - \Lambda(\hat{w}_{\text{on}}, \hat{\eta}^\pi), \text{ where}$$

$$\Lambda(\hat{w}_{\text{on}}, \hat{\eta}^\pi) = \frac{1}{2(T-k+1)} \sum_{t=0}^{T-k} \mathbb{E}\left[ |\hat{w}_{\text{on}}(S_{it}) - w_{\text{on}}(S_{it})| \right] + \sqrt{2L\gamma^k \, \mathbb{E}\left[ \bar{W}_1(\eta^\pi, \hat{\eta}^\pi) \right]}.$$

**Algorithm 1:** *CP for Infinite Horizon On-policy Evaluation*

---

**Data:** $\mathcal{D} = \{(S_{it}, A_{it}, R_{it}, S_{i,t+1}) : 1 \leq i \leq N, 1 \leq t \leq T\}$ and a test state $S_{\text{test}}$.

**Input:** $1 - \alpha$, target coverage level; $\mathcal{A}$, an on-policy distributional RL algorithm; $\mathcal{W}$, a density ratio estimation algorithm; $k$, step width; $B$, resampling number; $l$, subsample size; $\xi$, multiple subsampling parameter

**Output:** Prediction interval for $G^\pi(S_{\text{test}})$

1 Split the data: $\mathcal{D} = \mathcal{D}_{\text{tr}} \bigcup \mathcal{D}_{\text{cal}}$ where $\mathcal{D}_{\text{tr}} = \{(S_{it}, A_{it}, R_{it}, S_{i,t+1}) : (i,t) \in \mathcal{I}_{\text{tr}}\}$ and $\mathcal{D}_{\text{cal}} = \{(S_{it}, A_{it}, R_{it}, \ldots, S_{i,t+k}) : (i,t) \in \mathcal{I}_{\text{cal}}\}$. Here, $\mathcal{I}_{\text{tr}}$ and $\mathcal{I}_{\text{cal}}$ denote the indices of transitions in the training and calibration datasets, respectively.

2 Train a conditional return model $\hat{\eta}^\pi(s)$ using $\mathcal{A}$ based on $\mathcal{D}_{\text{tr}}$.

3 Obtain the value function estimator $\hat{v}^\pi(s)$, the expectation of $\hat{\eta}^\pi(s)$.

4 Obtain $\hat{w}_{\text{on}}(\mathbf{s})$ as an estimator of the density ratio (2) based on $\{S_{i0} : (i,0) \in \mathcal{I}_{\text{tr}}\}$ and $\{S_{it} : (i,t) \in \mathcal{I}_{\text{tr}}\}$ using $\mathcal{W}$.

5 **for** $b = 1 : B$ **do**

- Sample $l$ data tuples $\{(S_{it}, A_{i,t}, R_{i,t}, \ldots, S_{i,t+k}) : (i,t) \in \mathcal{I}_{\text{cal}}^{(b)}\}$ from $\mathcal{D}_{\text{cal}}$ according to the importance weight $\hat{w}_{\text{on}}(S_{it})$.

- Calculate pseudo return (1) and obtain $\widetilde{\mathcal{D}}_{\text{cal}}^{(b)} := \{(S_{it}, \widetilde{G}_{it}^{(k)}) : (i,t) \in \mathcal{I}_{\text{cal}}^{(b)}\}$.

- Calculate the nonconformity scores: $\{V_{it} := |\widetilde{G}_{it}^{(k)} - \hat{v}^\pi(S_{it})| : (i,t) \in \mathcal{I}_{\text{cal}}^{(b)}\}\}$.

- Obtain $\hat{q}_{1-\alpha\xi}^{(b)}$, the $\lceil l(1 - \alpha\xi) \rceil$-th smallest value of $\{V_{it} : (i,t) \in \mathcal{I}_{\text{cal}}^{(b)}\}$.

- Obtain $\widehat{C}_{N,\alpha\xi}^{(b)}(S_{\text{test}}) = \hat{v}^\pi(S_{\text{test}}) \pm \hat{q}_{1-\alpha\xi}^{(b)}$.

**Result:** A conformal predictive region for $G^\pi(S_{\text{test}})$ with a coverage rate of $1 - \alpha$ is

$$\widehat{C}_{N,\alpha}^{\text{on}}(S_{\text{test}}) = \left\{ G : \frac{1}{B} \sum_{b=1}^{B} I\left\{ G \in \widehat{C}_{N,\alpha\xi}^{(b)}(S_{\text{test}}) \right\} \geq 1 - \xi \right\}. \tag{5}$$

---

Theorem 1 shows that the deviation from nominal coverage depends on two main factors: (i) the estimation error in the importance weights, which arises due to the distribution shift, and (ii) the approximation error in the return distribution $\widehat{\eta}^\pi(s)$, measured by the Wasserstein distance. Notably, the second term decays with the truncation step $k$ at a rate proportional to $\gamma^k$. When the approximation error in the return distribution $\widehat{\eta}^\pi(s)$ is large, choosing a larger $k$ can help reduce the deviation from nominal coverage by relying more on observed rewards. However, this introduces a trade-off: if $k$ is too large, it becomes difficult to accurately estimate the off-policy weights, especially under substantial distributional shifts. In this case, the method effectively reduces to a Monte Carlo estimator that relies on full trajectories, resulting in the high variance we aim to avoid.

Next, we establish an asymptotic lower bound on the coverage of the PI in the off-policy setting. Let $\hat{w}_{\text{off}}(\cdot)$ be an estimate of the importance weight $w_{\text{off}}(\cdot)$ as defined in (4). Let $\widehat{C}_{N,\alpha}^{\text{off}}(\cdot)$ denote the conformal interval produced by Algorithm 1 in the supplementary material.

**Condition 2.** (i) The return distribution $\eta^\pi(s)$ has a Lebesgue density bounded by $L$ for all $s \in \mathcal{S}$. (ii) $\mathbb{E}[\hat{w}_{\text{off}}(\mathcal{H}_{t:t+k})|\mathcal{D}_{\text{tr}}] < \infty$, $\mathbb{E}[w_{\text{off}}(\mathcal{H}_{t:t+k})] < \infty$ for all $0 \leq t \leq T - k$, where $\mathcal{H}_{t:t+k} := (S_t, A_t, \ldots, S_{t+k})$ denotes the local trajectory segment following policy $\pi_b$, independent of $\mathcal{D}_{\text{tr}}$. (iii) (overlapping) $\pi_b(a|s)$ is uniformly bounded away from 0 for any $a, s$.

THEOREM 2 (**Off-Policy Coverage Guarantee**). *Assume Condition 2, and redefine* $\hat{w}_{\text{off}}(s_0, a_0, \ldots, s_{k+1})$ *as* $\hat{w}_{\text{off}}(s_0, a_0, \ldots, s_{k+1})/\frac{1}{T-k+1} \sum_{t=0}^{T-k} \mathbb{E}[\hat{w}_{\text{off}}(\mathcal{H}_{t:t+k})|\mathcal{D}_{\text{tr}}]$ *so that* $\frac{1}{T-k+1} \sum_{t=0}^{T-k} \mathbb{E}[\hat{w}_{\text{off}}(\mathcal{H}_{t:t+k})|\mathcal{D}_{\text{tr}}] = 1$. *Then we have*

$$\lim_{n \to \infty} \Pr\left( G^\pi(S_{\text{test}}) \in \widehat{C}_{N,\alpha}^{\text{off}}(S_{\text{test}}) \right) \geq 1 - \alpha - \Lambda(\hat{w}_{\text{off}}, \hat{\eta}^\pi), \text{ where}$$

$$\Lambda(\widehat{w}_{\text{off}}, \hat{\eta}^\pi) = \frac{1}{2(T - k + 1)} \sum_{t=0}^{T-k} \mathbb{E}\left[|\hat{w}_{\text{off}}(\mathcal{H}_{t:t+k}) - w_{\text{off}}(\mathcal{H}_{t:t+k})|\right] + \sqrt{2L\gamma^k \mathbb{E}\left[\bar{W}_1(\eta^\pi, \hat{\eta}^\pi)\right]}.$$

Theorem 2 shows that the coverage deviation has the same form as in the on-policy case (Theorem 1). The main difference is the additional estimation error in the importance weights $\widehat{w}_{\text{off}}$, which arises from evaluating a different target policy.

**Remark.** For continuous return distributions, the bounded Lebesgue density assumption is mild and typically satisfied in practice. It holds for many commonly-used distributions, including the Gaussian, exponential, and Gamma distributions with shape parameter no less than 1. For example, in Examples 1 and 2 of our experiments, the return distributions can be readily verified to satisfy this condition. In contrast, this assumption does not apply to discrete return distributions, as discrete random variables are not absolutely continuous with respect to the Lebesgue measure. Hence, the bounded density condition is neither required nor meaningful for discrete returns, as in Example 3 of our experiments.

## 5 Experiments

In this section, we conduct simulation studies to investigate the empirical performance of our proposed methods. In particular, we focus on the following two examples:

**Example 1: two-state MDP (Example 3 of [27])** The state space of the environment is discrete with two possible values: $x_1$ and $x_2$. The agent transfers from a current state to a different state with a certain probability determined by the policy and the discount factor is $\gamma = 0.8$. The reward obtained when transitioning from state $x_1$ is distributed as $N(2, 1)$, and the reward obtained when transitioning from state $x_2$ is distributed as $N(1, 1)$.

**Example 2: continuous state (Scenario B of [30])** The action is binary and $S_{t+1} = (S_{t+1,1}, S_{t+1,2})$, where $S_{t+1,1} = 3(2A_t - 1)S_{t,1}/4 + z_{t,1}$, $S_{t+1,2} = 3(1 - 2A_t)S_{t,2} + z_{t,2}$, $z_t = (z_{t,1}, z_{t,2})$, for $t \geq 0$, $\{z_t\}_{t \geq 0} \sim N(0_2, I_2/4)$ are i.i.d. and $S_0 \sim N(0_2, I_2)$. The immediate reward $R_t = 2S_{t+1,1} + S_{t+1,2} - (2A_t - 1)/4$. The discount factor is $\gamma = 0.8$.

For each example, we consider both an on-policy setting and an off-policy setting:

- In Example 1, when there is no policy shift, the probabilities of transferring from $x_1$ to $x_2$ and $x_2$ to $x_1$ are 0.4 and 0.8, respectively; when there exists a policy shift, the training data has the same transition dynamics as in the on-policy setting, while the test agent transitions from $x_1$ to $x_2$ with probability 0.5 and from $x_2$ to $x_1$ with probability 0.7.

- In Example 2, when there is no policy shift, both the observed data and the test agents satisfy $\Pr(A_t = 1|S_t) = 0.5\text{sigmoid}(S_{t,1}) + 0.5\text{sigmoid}(S_{t,2})$; when there exists a policy shift, the observed data follows the same policy as in the on-policy setting while the test data satisfies $\Pr(A_t = 1|S_t) = 0.6\text{sigmoid}(S_{t,1}) + 0.4\text{sigmoid}(S_{t,2})$.

**Implementation details.** The sample size is fixed to $N = 400$ for Example 1 and $N = 200$ for Example 2, with each trajectory consisting of $T = 30$ stages. For Example 1, we approximate the return distribution using 20 conditional quantiles estimated by QTD. In Example 2, where the state space is continuous, we use 30 conditional quantiles estimated by QTD and model the conditional quantile functions with a neural network. The detailed architecture of the neural network is provided in the supplementary material. We evaluate the performance of the proposed method with step sizes $k = 1, \ldots, 5$, and set the number of intervals $B = 50$. For each simulation, we generate 310 test points from the target policy to evaluate the converge probability. In the supplementary material, we include simulation results for Example 1 to examine the impact of $\xi$ and $k$, a comparison with [12] based on the same example, and an extension of Example 1 to a high-dimensional setting.

**Benchmark and Results.** We compare our method with the quantile region given by the learned QTD model (DRL-QR). Since the DRL algorithm directly learns the return distribution $\eta^\pi(S) := \mathcal{P}(G^\pi|S)$ by $\widehat{\eta}^\pi(S)$, a quantile region for the test instance $S_{\text{test}}$ can be constructed as $[\widehat{Q}_{a/2}(S_{\text{test}}), \widehat{Q}_{1-a/2}(S_{\text{test}})]$, where $\widehat{Q}_{\tilde{a}}(S_{\text{test}})$ is the $\tilde{a}$-th quantile of $\widehat{\eta}^\pi(S_{\text{test}})$. Figure 2 presents boxplots based on 50 independent repetitions. It shows that our method consistently achieves near-nominal 90% coverage across various $k$-step pseudo-returns in both on-policy and off-policy settings.

In contrast, the DRL-QR baseline suffers from undercoverage due to model bias in the estimated return distribution. This highlights the effectiveness of our conformal framework in correcting such bias and ensuring valid uncertainty quantification. We also observe that the average interval length increases with larger $k$, reflecting the higher variance introduced by longer truncation horizons.

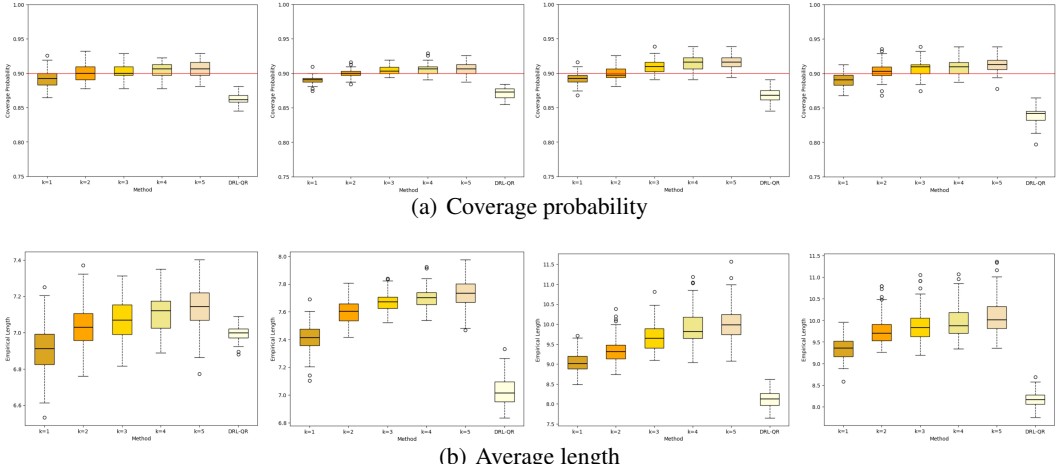

(a) Coverage probability

(b) Average length

Figure 2: Coverage probability and average interval length at the 90% level for the proposed method with $k$-step pseudo-returns ($k = 1, \ldots, 5$, from left to right) and DRL-QR (rightmost), under on-policy and off-policy settings in Example 1 (columns 1-2) and Example 2 (columns 3-4).

**Example 3: Mountain Car (adapted from [17])** We generate the dataset using a behavior policy defined as $\pi_b = a\pi_Q + (1 - a)\pi_U$, where $\pi_Q$ is a policy trained via Q-learning, $\pi_U$ is a uniformly random policy, and $a = 0.3$. The target policy is constructed similarly with $a = 0.2$, reflecting an off-policy setting. To conserve space, implementation details and results are provided in the supplementary material. As a benchmark, we apply kernel density estimation (KDE) to approximate the return distribution from Monte Carlo rollouts and construct baseline prediction intervals using quantiles (KDE-QR). As shown in Figure 1 of the supplementary material, our method effectively corrects the model bias in KDE and achieves near-nominal 90% coverage, highlighting the robustness of the proposed CP framework in a complex, continuous control task.

## 6 Conclusion

In this paper, we propose a novel CP framework for infinite-horizon policy evaluation with asymptotic coverage guarantees. By constructing $k$-step pseudo-returns, our method balances predictive accuracy and statistical efficiency, addressing key challenges in long-horizon evaluation. This formulation enables the construction of valid PIs without relying on full trajectory rollouts. Although the choice of $k$ remains underexplored, we suggest practical remedies such as evaluating stability across multiple $k$ values (e.g., $k = 1, \ldots, 5$) or aggregating PIs across different $k$. Since these intervals are correlated, aggregation is nontrivial. A promising direction is to construct a unified prediction region by combining the corresponding p-values, leveraging the connection between prediction intervals and hypothesis testing. Methods such as the Cauchy Combination Test [20], which are robust to arbitrary dependencies, offer a viable approach. Moreover, extending our framework to policy optimization represents an exciting avenue for future work and could further broaden the applicability of conformal prediction in RL.

## Acknowledgement

Zhang's research was supported by the National Natural Science Foundation of China (Grant No. 12471280) and the Shanghai Municipal Education Commission (Grant No. 2024AI01002). Liu's research was supported by the National Natural Science Foundation of China (Grant No. 12571283).

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
