# OpenReview forum: "Conformal Prediction Beyond the Horizon: Distribution-Free Inference for Policy Evaluation"
_NeurIPS.cc/2025/Conference — NeurIPS 2025 poster_

### Official Review · Reviewer_jGpc · 2025-06-04

**Clarity:** 3
**Significance:** 3
**Originality:** 3
**Rating:** 5
**Confidence:** 4

**Summary:**

This paper leverages conformal prediction to derive distribution-free predictive intervals (PI) for the return $G^\pi(s_{test})$ in discounted MDPs for a fixed policy $\pi$ and a test state $s_{test}$, applicable to both on-policy and off-policy settings. The method is particularly suited for neural network-based predictions due to its distribution-free nature. Specifically, the authors split the data into training and calibration sets: the QTD algorithm is trained on the training set to estimate the value function and generate approximate samples of $G^\pi(s)$, while k-step TD estimates ($G^\pi(s_k)$ is sampled from the QTD model) of $G^\pi(s_{cal})$ on the calibration set are used to compute nonconformity scores (defined as the difference between the k-step TD estimates and the value function estimates). To enhance performance, the authors incorporate weighted subsampling and resampling techniques.
Under the assumption that $G^\pi(s)$ admits a bounded Lebesgue density, the authors establish coverage guarantees for the derived PI. Numerical experiments demonstrate that, compared to PI obtained directly via distributional RL (DRL), the proposed method achieves the desired coverage probability without significantly inflating interval widths.

**Questions:**

The coverage guarantees rely on the return distribution having a bounded Lebesgue density. How does the method perform empirically when this assumption is violated which is often the case? A brief discussion or experiment would strengthen the practical relevance.

**Ethical Concerns:**

["NO or VERY MINOR ethics concerns only"]

**Final Justification:**

The authors have addressed my concerns. I will keep my positive score unchanged.

**Limitations:**

Yes

**Quality:**

3

**Strengths And Weaknesses:**

Strengths:

This work elegantly integrates conformal prediction with distributional RL and replay buffer techniques to derive a theoretically guaranteed predictive interval for $G^\pi(s_{test})$. The empirical results convincingly validate the method’s effectiveness, making this a compelling contribution.

Weaknesses:

I do not find major flaws, I offer the following refinements to enhance clarity and rigor.

1. Notational Clarity: The symbol m is overloaded for both QTD and resampling parameters.

2. Clarify Methodology for Off-Policy Evaluation (Line 171-178): The current description of estimating $\eta^\pi(s,a)$ in the off-policy setting is vague. Explicitly state the distributional off-policy evaluation method used in the paper (e.g., distributional variant of Fitted Q Evaluation) or cite relevant work.

3. Clarify Methodology used in Eqn. (4): Please explain how to estimate P(\delta | s) (e.g. logistic regression?) and explicitly note that
 P(δ=0)/P(δ=1) can be estimated via frequency ratios (if this aligns with the authors’ intent).

4. In Algorithm 1, on Result line, conformal predictive region can be replaced with conformal predictive interval (because it is indeed an interval). And the impact of the multiple subsampling parameter $\xi$ is not discussed in the paper (although it is discussed in the Appendix).

---

> ### Author Rebuttal · Authors · 2025-07-31
>
> # Author response to reviewer jGpc
>
> We sincerely thank Reviewer jGpc for the recognition of our work and for providing constructive comments.
>
> ## Q1: Notational Clarity.
>
> Thank you for your careful reading and thoughtful comments. To avoid confusion, we have revised the notation in Algorithm 1: the subsample size, previously denoted by $m$ in the submitted version, is now written as $l$ in the revised manuscript.
>
> ## Q2: Clarify Methodology for Off-Policy Evaluation (Line 171-178).
>
> Thank you for your valuable comments. Lines 171–178 aim to explain how the QTD algorithm of [3] can be adapted to the off-policy setting.
> The QTD algorithm presented in Algorithm 1 of [3] is designed for the **on-policy** setting. Extending QTD to the **off-policy** setting requires careful modifications to account for the distribution shifts between the behavior policy used to collect data and the target policy.
>
> - In the on-policy case, QTD estimates
>   the return distribution conditioned on the initial state, $\widehat{\eta}^{\pi}(s)$, using the following iterative update:
>   $$
>   \theta(s,i) \leftarrow \theta(s,i)+\alpha\frac{1}{m}\sum_{j=1}^{m}\left[\tau_i - I(r+\gamma\theta(s^\prime,j)-\theta(s,i)<0) \right],
>   $$
>   where $\theta(s, i)$ denotes the $i$-th quantile of $\widehat{\eta}^{\pi}(s)$ and $(s, a, r, s')$ is sampled under the behavior policy $\pi$, which in the on-policy case coincides with the target policy.
>
> - In contrast, the off-policy setting requires
>   estimating the return distribution conditional on both the state and the action, $\widehat{\eta}^{\pi}(s, a)$, using the modified update:
>   $$
>   \theta(s,a,i) \leftarrow \theta(s,a,i)+\alpha\frac{1}{m}\sum_{j=1}^{m}\left[\tau_i - I(r+\gamma\theta(s^\prime,a^\prime,j)-\theta(s,a,i)<0) \right],
>   $$
>   where $\theta(s, a, i)$ is the $i$-th quantile of $\widehat{\eta}^{\pi}(s, a)$, and $(s, a, r, s')$ is sampled from the behavior policy $\pi_b$. Crucially, $a'$ is drawn from the target policy $\pi$, and the result is marginalized over the action space according to $\pi$. This added complexity arises from the need to correct for the action distribution mismatch between the behavior and target policies.
>
> - This distinction is analogous to the difference
>   between **the TD algorithm** for estimating the state-value function $V^{\pi}(s)$ and **the SARSA algorithm** for estimating the state-action value function $Q^{\pi}(s,a)$. We will clarify this point in the revised paper and refer readers to relevant literature, including [1] and [2] for more details on distributional off-policy evaluation.
>
> ## Q3: Clarify Methodology used in Eqn. (4).
>
> Thank you very much for the insightful comments. We estimate $P(\delta \mid s)$ using logistic regression in our implementation, although other models such as neural networks can also be employed. As for the ratio $P(\delta = 0)/P(\delta = 1)$, it is a constant and does not need to be explicitly estimated, since we normalize the resulting weights into probabilities in the subsequent weighted subsampling procedure. We will clarify this point in the revised paper.
>
> ## Q4: In Algorithm 1, conformal predictive region can be replaced with interval. And the impact of $\xi$ is not discussed in the paper (although it is discussed in the Appendix).
>
> Thank you for your thoughtful comment. We will replace the conformal prediction region in result line of Algorithm 1 by conformal prediction interval. In addition, following your suggestion, we conducted experiments with $\xi=0,1,0.2,\ldots,0.9$ in Example 1, as shown in **Rebuttal Table C**. The results indicate that smaller $\xi$ and larger $k$ tend to produce over-coverage, while overall, settings with $\xi\ge0.5$ and $k=2,3$ yield satisfactory performance. We will incorporate these findings into the paper.
>
> **Rebuttal Table C.**  We conduct the experiments for Example 1 across $xi$ for $k=2,3,4$. Each experiment is repeated 100 times, and we report the mean and standard deviation of coverage probability (cov) and interval length (len) at the nominal coverage rate 90%.
>
> |||||||||
> |-|-|-|-|-|-|-|-|
> |**on policy**|||||||
> |cov|k=2|k=3|k=4|len|k=2|k=3|k=4|
> |$\xi$=0.1|0.95(0.01)|0.96(0.01)|0.96(0.01)| |10.21(0.20)|10.71(0.21)|11.04(0.26)|
> |$\xi$=0.2|0.95(0.01)|0.95(0.01)|0.95(0.01)| |9.68(0.15)|10.10(0.16)|10.40(0.17)|
> |$\xi$=0.3|0.94(0.01)|0.95(0.01)|0.95(0.01)| |9.30(0.12)|9.67(0.14)|9.95(0.16)|
> |$\xi$=0.4|0.92(0.01)|0.94(0.01)|0.95(0.01)| |8.98(0.10)|9.34(0.13)|9.62(0.15)|
> |$\xi$=0.5|0.92(0.01)|0.93(0.01)|0.94(0.01)| |8.73(0.08)|9.07(0.13)|9.33(0.15)|
> |$\xi$=0.6|0.91(0.01)|0.92(0.01)|0.93(0.01)| |8.53(0.09)|8.87(0.13)|9.09(0.16)|
> |$\xi$=0.7|0.91(0.01)|0.92(0.01)|0.92(0.01)| |8.37(0.09)|8.69(0.12)|8.92(0.14)|
> |$\xi$=0.8|0.90(0.01)|0.91(0.01)|0.92(0.01)| |8.24(0.10)|8.56(0.13)|8.78(0.14)|
> |$\xi$=0.9|0.90(0.01)|0.91(0.01)|0.92(0.01)| |8.20(0.12)|8.51(0.14)|8.72(0.16)|
> |**off policy**|||||||
> |cov|k=2|k=3|k=4|len|k=2|k=3|k=4|
> |$\xi$=0.1|0.96(0.01)|0.96(0.01)|0.97(0.01)| | 10.17(0.19)|10.68(0.22)|10.95(0.30)|
> |$\xi$=0.2|0.95(0.01)|0.95(0.01)|0.96(0.01)| |9.62(0.15)|10.08(0.17)|10.33(0.20)|
> |$\xi$=0.3|0.94(0.01)|0.95(0.01)|0.96(0.01)| |9.24(0.12)|9.64(0.15)|9.90(0.17)|
> |$\xi$=0.4|0.93(0.01)|0.94(0.01)|0.95(0.02)| |8.93(0.10)|9.30(0.13)|9.57(0.15)|
> |$\xi$=0.5|0.92(0.01)|0.93(0.01)|0.94(0.01)| |8.68(0.11)|9.03(0.14)|9.28(0.15)|
> |$\xi$=0.6|0.92(0.01)|0.93(0.01)|0.93(0.01)| |8.43(0.11)|8.78(0.14)|8.99((0.14)|
> |$\xi$=0.7|0.91(0.01)|0.92(0.01)|0.93(0.01)| |8.26(0.11)|8.61(0.13)|8.82(0.14)|
> |$\xi$=0.8|0.91(0.01)|0.92(0.01)|0.92(0.01)| |8.13(0.11)|8.47(0.14)|8.67(0.14)|
> |$\xi$=0.9|0.91(0.01)|0.92(0.01)|0.92(0.01)| |8.07(0.13)|8.41(0.16)|8.61(0.15)|
>
> ## Q5:  A brief discussion or experiment on  bounded Lebesgue density would strengthen the practical relevance.
>
> We appreciate your constructive comments. Our response is structured in three parts:
>
> - The assumption of a bounded Lebesgue density applies only to continuous return distributions, since discrete random variables are not absolutely continuous with respect to the Lebesgue measure. Accordingly, this condition is not required or applicable when the return is discrete, as in Example 3 of our experiments. We acknowledge this and will clarify it in the revised paper.
>
> - For continuous return distributions, the bounded density assumption is mild and satisfied by many commonly used distributions such as Gaussian, exponential, and Gamma with shape parameter no less than 1. For instance, in Examples 1 and 2 of our experiments, the return is continuous and satisfies the bounded density condition.
>
> - To illustrate, consider **Example 1** in our paper. In this example, the state space of the environment is discrete with two possible values: $S_t=1$ and $S_t=2$. The agent transfers from a current state to a different state with a certain probability determined by the policy and the discount factor is $\gamma = 0.8$.
> The reward obtained when transitioning from state $1$ is distributed as $N(2, 1)$, and the reward obtained when transitioning from state $2$ is distributed as N (1, 1), i.e., $R_t|S_t\sim N(-3S_t+5,1)$.
> Given a history of states $(S_0,S_1,\ldots)$, the $R_t$'s are independent Gaussian variables, so the return $G=\sum_{t=0}^\infty\gamma^tR_t$ also follows a Gaussian distribution.
> Please note that the Gaussian density is upper-bounded by $1/\sqrt{2\pi\sigma^2}$ where $\sigma^2$ is its variance.
> The variance of $G$ conditional on $(S_0,S_1,\ldots)$ is $\sum_{t=0}^\infty \gamma^{2t} = 1/(1-\gamma^2)$, so the conditional density is upper-bounded by $\sqrt{\frac{1-\gamma^2}{2\pi}}$. Since this upper bound is irrelevant with $(S_1,S_2,\ldots)$, the conditional density of $G$ given $S_0$ is also bounded by $\sqrt{\frac{1-\gamma^2}{2\pi}}$.
> Similarly, we can verify that **Example 2** also satisfies this bounded density condition.
>
>
> # Reference
>
> [1] Distributional Off-policy Evaluation with Bellman Residual Minimization.
>
> [2] Distributional Off-Policy Evaluation in Reinforcement Learning.
>
> [3] An analysis of quantile temporal-difference learning.

---

> > ### Comment · Reviewer_jGpc · 2025-08-03
> >
> > Thank you for your detailed reply, which has addressed my concerns. I will keep my score.

---

> > > ### Author Response · Authors · 2025-08-03
> > >
> > > Dear Reviewer jGpc,
> > >
> > > Thank you for your feedback. We are pleased to hear that our responses have addressed your concerns and that you maintain your score. We truly appreciate your time and support.
> > >
> > > Best, Authors

---

### Official Review · Reviewer_SpYL · 2025-06-29

**Clarity:** 2
**Significance:** 3
**Originality:** 2
**Rating:** 4
**Confidence:** 3

**Summary:**

The paper  proposes a split-conformal‐prediction for infinite-horizon reinforcement-learning (RL) returns. The authors propose a method based on distributional RL to compute pseudo-returns, which are then used to build the conformal scores. For their method, they derive an asymptotic guarantee on coverage.
    Experiments on a two-state toy MDP, a continuous 2-D benchmark and Mountain-Car show that the method hits the nominal 90 % coverage while a plain DRL quantile interval under-covers.

**Questions:**

- Why use the term "Prediction interval" instead of "Confidence interval"? Furthermore, when writing PI it looks like Policy Improvement.
- Eq. 2: what if $k$ exceeds the trajectory length?
- Possible misunderstanding: why it seems like that a small value of $k$ seems to improve the coverage? the second term in the square root will be larger (thm 1)
- Clarify $N$ and $n$ in the manuscript.
- Unclear what is ${\cal I}$ in the algorithm

Possible typos:
- Line 16: "As reinforcement learning (RL) are " -> "As reinforcement learning (RL) _algorithms_ are"?
- Line 93: "uniform length" -> "_constant_ length"
- Figure 1, under conformal prediction, (1) is repeated twice

**Ethical Concerns:**

["NO or VERY MINOR ethics concerns only"]

**Final Justification:**

The authors’ responses substantively address my main concerns about correctness and positioning, thus I increase my score from 3 to 4.

The authors have  substantively address my main concerns about correctness and positioning.  They also added a comparison to Foffano et al. (2023),  and discussed the $k$-step trade-off and why WCP can be unstable in long-horizon settings. These clarifications,tip the balance for me toward acceptance, albeit narrowly. Remaining issues regarding presentation of the concepts are camera-ready items rather than fatal flaws.


Residual weaknesses:
-   Guarantees remain asymptotic; the role of replay/weighted subsampling in restoring exchangeability should be stated crisply in the main text, with assumptions up front.
- Evaluation is still mostly low-dimensional; at least two higher-dimensional or continuous-control benchmark would improve external validity. Off-policy experiments should be given the priority given the high practical relevance.

Provided the authors address all the concerns that were initially raised in my review, I suggest a borderline accept score.

**Limitations:**

Limitations and impact do not seem to be discussed.

**Paper Formatting Concerns:**

I didn't notice any major concern.

**Quality:**

2

**Strengths And Weaknesses:**

**Strenghts**
- Using an approach based on computing the pseudo-returns make more intuitive sense than prior approaches, as well as using a replay buffer for breaking dependency.
- Experiments seem to demonstrate the validity of the approach
- The authors provide an asymptotic coverage analysis for their method


**Weaknessess**

- Problematic claims:
  - The authors in the introduction claim that "Unobserved Returns" is a challenge, since full return is not directly observable in the infinite-horizon setting. However, that depends also on the characteristics of the underlying MDP. In some MDPs (e.g., finite and communicating), if one can estimate well the stationary distribution, then it is not necessary to observe the "entire" return.
  - It's not clear why the authors state that "Model Misspecification" is a challenge for conformal prediction, since it should be a model-free approach. The overall statement is a bit questionable.
  - Confusion with DRL in line 125. When the authors write "DRL-based quantile intervals, referred to as DRL-QR,", in reality this type of construction is not only limited to DRL, but it's more general than that. It is quite common to use empirical CDFs to build confidence intervals using the lower/upper quantiles (e.g., using bootstrap).
  - About not using WCP the authors write "in the RL setting, intermediate transitions along the test trajectory are not available at prediction time, making direct application of WCP infeasible". But one could use a learned model of the transition, or use an approach base don eq.14 of Foffano et al. 2023
  - Line 235: not clear why the authors write " Previous studies have bounded this gap using total variation distance, which235
fails to capture how different choices of k in k-step rollouts affect the coverage gap. ". Isn't their work the first to use k-step rollouts? Why TV  fails?

- Missing assumptions, notation and possibly not correct equations
  - The MDP model is missing. The authors write "we use the standard RL framework", but what is this standard framework? What are the assumptions on the MDP? Furthemore, also the assumptions on how the trajectories are collected are missing.
  - The authors define $\mathcal{P}$ to be the transition function. What is then ${\rm d}{\cal P}_0(s)$ in eq 4? Similarly, what do they mean by $P$ in eq 4? They write " is the distribution of test state", but a precise definition is missing. Overall, the equation does not seem to be correct.
  - Off policy setting, lines 171-178: this paragraph just does not seem correct. The goal is to estimate the return distribution from data collected from a different policy, but the section does not seem to explain that (not even uses importance sampling), and I am confused about the intent of the authors here overall.
  - Theorem 1/2: I don't think this theorem is correct, due to the dubious definition of ${\rm d}{\cal P}_0(s)$  (similarly for the calibration) in eq 4. This should depend on the stationary distributions (and we need assumptions on the MDP). Furthermore, what is the limit in $n$ (you mean $N$?) and under which assumptions on the MDP/data/policies we have that the Lebesgue density is bounded by $L$? Also, the boundedness on the importance sampling terms puts assumptions on the data collected, this should be stated more clearly. Also, the statement only holds asymptotically (as the number of trajectories tends to infinity), which affects the strength of the result.
  - From the proofs is also not clear how the replay buffer should make the samples exchangeable.
- Experiments are limited to simple examples, and only compare to DRL, but not to other techniques (e.g., Foffano et al 2023).
- I believe the main challenge in conformal prediction for MDPs is to adequately address the off-policy setting, but as of now,  I believe the paper lacks sufficient clarity/precision to address this point.

- Missing reference on conformal prediction with Markovian data:  Zheng, Frédéric, and Alexandre Proutiere. "Conformal Predictions under Markovian Data." Forty-first International Conference on Machine Learning.

---

> ### Author Rebuttal · Authors · 2025-07-31
>
> # Author response to reviewer SpYL
>
> We thank Reviewer SpYL for your careful reading and valuable comments.
>
> ## Q1: Problematic claims
> - Q1-1:  We agree that in certain classes of MDPs the stationary distribution $\mu^\pi(s,a)$ can be well estimated. However, our method depends critically on a more challenging quantity: the long-term return distribution
>    $$\eta^{\pi}(s) = \mathrm{Law} \left( \sum_{t=0}^{\infty} \gamma^t r_t \middle| s_0 = s \right).$$
>    Estimating this distribution (e.g., quantiles) requires knowledge **not only** of $\mu^{\pi}$ but also of the conditional distribution of cumulative future rewards. For a detailed background, please see DRL literature such as [1].
>
>    In infinite-horizon settings, the return is inherently unobservable from finite trajectories, especially under off-policy data, which motivates our work.
>
>    While long-horizon trajectories enable Monte Carlo estimation, it is more common to exploit time-homogeneity of MDP to use all immediate transitions, as in temporal difference (TD) methods, which is our focus in this work.
> - Q1-2: The term “model misspecification” refers to the difficulty of constructing valid PIs using estimated return distributions. For example, quantiles estimated by DRL models may be inaccurate, leading to poor coverage.
>
>    While we agree that CP is model-free with guaranteed marginal coverage, applying CP to infinite-horizon and continuous RL remains challenging. Our work addresses this gap, and we will clarify this point in the paper.
> - Q1-3: While quantile-based intervals via empirical CDFs are standard, in infinite-horizon RL, full return trajectories are rarely observed. Thus, direct estimation is often infeasible. Though long-horizon Monte Carlo rollouts can approximate return distributions, they are often sample-inefficient, especially in off-policy or high-variance settings. Instead, like much of RL, our work adopts the TD-based approach, where return quantiles are learned from immediate transitions. “DRL-QR” refers to intervals built from these TD-based quantile estimates.
> - Q1-4: WCP typically assumes access to full test-time covariates, while our setting only observes the initial state $S_{\text{test},0}$ at test time. Estimating the WCP weight from Foffano et al. (2023, Eq.12) requires **marginalizing over full trajectories**, which are unobserved and intractable to approximate without strong assumptions.
>
>   We may estimate the WCP weight by solving the objective function in their Eq.14, this requires additional modeling assumptions and may suffer from high variance, particularly in long-horizon settings. The resulting weights can be unstable and sensitive to model choice, limiting practical use in our setting.
>
>   Finally, simulating trajectories with learned **transition model** $p(s' \mid s,a)$ introduces approximation error and places the burden of accuracy on the model class. **More critically**, it doesn’t address a key requirement of conformal prediction: distributional alignment between calibration and test data.
> - Q1-5: We initially considered a TV-based coverage bound following [5], but it fails to capture how rollout length $k$ affects distributional mismatch. Inspired by [10] which shows Wasserstein distance better reflects conformal coverage under distribution shift (see their Section B for a detailed comparison), we adopt a Wasserstein-based bound to characterize the effect of $k$.
> ## Q2: Missing assumptions and so on
> - Q2-1: We assume a standard time-homogeneous Markov Decision Process with data collected from i.i.d. trajectories under a fixed behavior policy. We originally stated these assumptions in the Supplement and  will move them to the main paper.
> - Q2-2: In our notation, $\mathcal{P}$ denotes a probability distribution. For example, $\mathcal{P}_0$ represents the distribution of the initial state. Since we only observe the initial state of the test trajectory, we refer to this as the distribution of the test state. We use $P$ to denote the probability measure induced by a distribution.
> - Q2-3: Lines 171–178 explain how the QTD algorithm from [7], originally developed for the on-policy case, is adapted to the off-policy setting in our work. The key change is that we estimate the return distribution $\widehat{\eta}^\pi(s,a)$ instead of $\widehat{\eta}^\pi(s)$, and account for the mismatch between the behavior policy $\pi_b$ and the target policy $\pi$ by sampling $a'$ from $\pi$ and marginalizing accordingly. This is conceptually similar to moving from TD (estimating $V^\pi(s)$) to SARSA (estimating $Q^\pi(s,a)$). For full update rules, please see our response to  Q2 of Reviewer #4. We will  clarify this point and cite [4,8] for related work on off-policy distributional evaluation.
> - Q2-4: Our responses are organized point-by-point below.
>
>    1. **$\mathcal{P}_0$ and assumptions.** Please refer to Q2-1 and Q2-2.
>
>    2. **On $n$.** Please refer to Q4-4.
>
>    3. **Boundedness of density.** The assumption that the Lebesgue density is bounded holds for any continuous random variable that has a continuous density function that does not tend to infinity at the boundary of its support. Examples include Gaussian distribution, exponential distribution, etc. It can be verified that both   **Example 1** and  **Example 2** satisfy this assumption. Please refer to our response to Q5 of Reviewer #4 for details.
>
>    4. **Assumptions of IS.** We interpret the comment as referring to the weighting function $w$. Our analysis only requires a bounded **first moment** of $w$, which is a common and mild assumption in the CP literature, including [5,11].
>    5. **Asymptotic coverage.** The theorem ensures asymptotic coverage as the number of calibration samples $n$ tends to infinity, **not the number of trajectories**. Please refer to Q4-4.
> - Q2-5: To address the issue of temporal dependence among transitions, we follow a common technique in the RL literature known as experience replay ([6,7]). Specifically, we apply weighted subsampling from a large replay buffer to construct a subset of transitions that are approximately i.i.d. This approach is widely used in RL to break temporal correlations and is supported both empirically and theoretically in prior work such as [2].
> ## Q3: Weekness 3-5
> - Q3-1: **Weekness 3.** Following your suggestion, we add a comparison with Foffano et al. (2023) in **Rebuttal Table A**, where our method demonstrates superior performance across coverage probability and average length.
>
>    We highlight that Foffano et al.(2023) address **finite-horizon** MDPs and mitigate distributional shift via importance weighting (see their Eq.12), but their method is subject to the curse of horizon. In contrast, our approach targets the infinite-horizon setting and reduces the challenge to a $k$-step distributional shift problem. Please refer to our response to Q1 of Reviewer #1 for further discussion.
>
>   **Rebuttal Table A.** We compare the performance on Example 1 with a fixed horizon of 20. For Foffano’s method, we use their gradient-based linear regression to train $w(x, y)$ and apply WCP. Each experiment is repeated 100 times, and we report the mean and standard deviation of coverage probability (cov) and interval length (len) at the nominal coverage rate 90%.
> ||||
> |-|-|-|
> ||cov|len|
> |Foffano|0.85(0.01)|6.51(0.11)|
> |our proposal|||
> |k=2,$\xi$=0.5|0.91(0.01)|7.67(0.15)|
> |k=3,$\xi$=0.5|0.91(0.01)|7.77(0.13)|
> |k=2,$\xi$=0.6|0.92(0.01)|8.00(0.16)|
> |k=3,$\xi$=0.6|0.92(0.01)|8.02(0.15)|
>
> - Q3-2: **Weekness 4.**  We agree that off-policy evaluation poses major challenges for applying CP to MDPs. Addressing this is a central contribution of our work. To mitigate severe distributional shift in the infinite-horizon setting, we estimate the return distribution using DRL and apply CP to correct model misspecification. We also propose $k$-step pseudo returns to reduce the infinite-horizon shift to a more tractable $k$-step shift.
> - Q3-3: **Weekness 5.** Thanks for pointing out this interesting paper. We will cite it.
> ## Q4: Questions 1-5
> - Q4-1:  A confidence interval is an interval estimate for a parameter (fixed), whereas a prediction interval is an interval estimate for a random variable (random)  with a large probability. Given an initial state of a trajectory, the return is unobserved and can be seen as a random variable.
> This is why we use “Prediction interval” instead of “Confidence interval”.
> “PI” is defined at Line 23 and follows standard terminology in CP.
> - Q4-2: The event that $k$ exceeds the trajectory length will never happen as we are focusing on the  infinite-horizon settings.
> - Q4-3: We apologize for the typo in $\Lambda$ in Theorems 1/2. It should be the sum of two estimation errors and we will correct it.
> - Q4-4: As defined in Section 2,  $N$ is the number of trajectories and $T$ the horizon. In line 242, $n$ denotes the number of calibration samples, not trajectories. Since we extract overlapping $k$-step transitions, $n \approx NT/2$ in split CP.
> - Q4-5: In Algorithm 1, $\mathcal{I}$ refers to the index of the transition $(S_{i,t}, A_{i,t}, R_{i,t}, S_{i,t+1})$ in the training dataset. We will add this definition.
> ## Q5: Possible typos
>
> We will correct these typos in our paper.
>
> [1] A distributional perspective on reinforcement learning.
>
> [2] A theoretical analysis of deep Q-learning.
>
> [3] Conformal off-policy evaluation in markov decision processes.
>
> [4] Distributional Off-policy Evaluation with Bellman Residual Minimization.
>
> [5] Conformal inference of counterfactuals and individual treatment effects.
>
> [6] Self-improving reactive agents based on reinforcement learning, planning and teaching.
>
> [7] Human-level control through deep reinforcement learning.
>
> [8] Distributional Off-Policy Evaluation in Reinforcement Learning.
>
> [9] An analysis of quantile temporal-difference learning.
>
> [10] Wasserstein-regularized conformal prediction under general distribution shift.
>
> [11] Conformal off-policy prediction.

---

> > ### Comment · Reviewer_SpYL · 2025-08-04
> >
> > I thank the authors for their response and for the additional comparison, that is appreciated!
> >
> > Regarding my concerns, I advise the authors to avoid making claims such as "In infinite-horizon settings, the return is inherently unobservable from finite trajectories" because it is ambiguous (what does it mean unobservable? why is is inherently unobservable?). Moreover, in the end the fact that we don't have infinite trajectories is usually not a problem, since we exploit the fact that the discount is smaller than 1. Overall, I advise authors to improve in clarity in their next iteration of the manuscript.
> >
> > Can also the authors better address some of my concerns, specifically:
> >
> > - Regarding eq. 4, where is $P$ defined exactly? and why the r.h.s of 4 should hold?
> >
> > - Can the authors point out in the proofs where do you use the following "we apply weighted subsampling from a large replay buffer to construct a subset of transitions that are approximately i.i.d. "? Where do I see that?
> >
> > - What are the challenges in obtaining non-asymptotic guarantees?

---

> > > ### Author Response · Authors · 2025-08-05
> > >
> > > We sincerely appreciate the reviewer’s feedback. Below, we address the further concerns in detail.
> > > ## Q1: Clarify "inherently unobservable"
> > > Sorry for the unclear description.  The goal of this paper is to construct PIs for returns in the infinite-horizon MDP setting, where the return is defined as $G  = R_0 + \gamma R_1 + ... + \gamma^{T-1} R_T + ... $ with $\gamma \in [0,1)$. However, in practice we can only observe finite-horizon trajectories, yielding a truncated return $G_T = R_0 + \gamma R_1 + ... + \gamma^{T-1} R_T $. Since future rewards beyond $T$ are unobserved, the difference $G-G_T$ is inherently unobservable, making $G$ itself unobservable from finite data.
> > >
> > > We agree that discounting ($\gamma < 1$) makes the truncation error $G-G_T$ small as $T$ increases. However, **in offline RL**, when $T$ is moderate, the error is still non-negligible.
> > >
> > > Finally, we highlight that our method leverages both the Markovian and time-homogeneous structure of MDPs. These properties allow us to **reuse observed transitions across different time steps**. For this purpose, when we rely only on immediate transitions of the form $(S_{t},A_{t},R_{t}, S_{t+1},A_{t+1},..., S_{t+k}, A_{t+k}, R_{t+k}, S_{t+k+1})$, the unobserved portion of the return beyond the $k$ observed steps is non-negligible as $k$ is small in general.
> > >
> > > We will revise the paper to clarify these points regarding the "unobserved return" argument in lines 39–40.
> > >
> > > ## Q2: Why Eq 4 holds
> > > Suppose that we have $n_1$ i.i.d. calibration states $\{s\_{11}, \ldots, s\_{1n_1}\}\sim\mathcal{P}\_{cal}$ and  $n_0$ i.i.d. test stats $\{s\_{01}, \ldots, s\_{0n_0}\}\sim\mathcal{P}_{0}$. We aim to estimate the ratio  $w(s) = d\mathcal{P}_0(s)/\mathcal{P}\_{cal}(s)$. Let $n=n_0+n_1$  and $\eta = n_0/n$. For each $s\_{0i}$ ($1\leq i\leq n_0$), we define an indicator $\delta\_{0i} = 1$. Similarly, for each $s\_{1i}$ ($1\leq i\leq n_1$), we define an indicator $\delta\_{1i} = 0$. Now we pool  the observed data  $\{(s\_{0i}, \delta\_{0i}): 1\leq i\leq n_0 \}$ and $\{(s\_{1i}, \delta\_{1i}): 1\leq i\leq n_1 \}$ together and denote the resulting dataset as  $\{(s_i, \delta_i):  1\leq i\leq n\}$.  Assuming  $n_0$ and $n_1$ are fixed integers, we use $\mathcal{P}$ to denote the joint distribution of $(s, \delta)$ in the pool dataset, or the expectation of the empirical distribution of the pool data. It can be found that
> > > $$\mathcal{P}(s, d) = \eta \mathcal{P}\_{cal}(s) I(0\leq d) + (1-\eta) \mathcal{P}_0(s) I(1\leq d).$$
> > >
> > > Let  $P$ denote the usual probability operator, with $P(s)$  a density function of $S$ and $P(\delta=1)$   a probability. It follows that   $\mathcal{P}_0(s) = P(s|\delta=1)$ and $\mathcal{P}\_{cal}(s) = P(s|\delta=0)$. Using Bayes’ rule, we have:
> > > $$
> > > \frac{\mathcal{P}_0(s) }{ \mathcal{P}\_{cal}(s) }
> > > =  \frac{P(s|\delta=1)}{P(s|\delta=0)}
> > >  = \frac{P(\delta = 1|s)  P(s)/ P(\delta = 1) }{ P(\delta = 0|s)  P( s)/P(\delta = 0)}
> > > =\frac{P(\delta = 1|s)   }{ P(\delta = 0|s) }
> > > \times \frac{  P(\delta = 0) }{  P(\delta = 1)}.
> > > $$
> > > This is exactly the r.h.s of Eq. 4. We will define $P$ and clarify this following Eq. 4.
> > >
> > > ## Q3: Clarify in proof
> > > Theoretically, the calibration set used in CP is a subsample drawn from the replay buffer and thus follows a (weighted) empirical distribution defined in lines 68–69 for the on-policy setting and lines 133–134 for the off-policy setting in the Supp. When the data points are (approximately) i.i.d., the empirical distribution converges to the true distribution by the law of large numbers (see also [1]).
> > >
> > > Due to the discrepancy between the empirical distribution and the true distribution, the theoretical coverage can only attain the nominal level asymptotically, as the sample size tends to infinity. In our analysis, the terms $M_{21}$ (lines 85–92) and $\widetilde{M}_{21}$ (lines 147–156) in the Supp explicitly characterize the coverage gap induced by this deviation.
> > >
> > > We emphasize that the idea of weighted subsampling is inspired by the experience replay mechanism, which is a widely adopted and well-established practical method. In fact, some prior works directly assume that the resampled tuples are i.i.d.; see [2].
> > >
> > > ## Q4: Challenges for non-asymptotic guarantees
> > > The asymptotic coverage guarantee is mainly due to subsampling, which is standard in the CP literature [3]; see also Q3. While finite-sample coverage could be derived by using only the initial part of each trajectory (avoiding subsampling), this reduces to a trajectory-wise method and discards valuable information from replay buffer. Empirically, this leads to less efficient prediction intervals. We instead prioritize leveraging the MDP’s time-homogeneity  and replay buffer to improve statistical efficiency.
> > >
> > > [1] Week dependence with examples and applications.
> > >
> > > [2] A theoretical analysis of deep Q-learning.
> > >
> > > [3] Distribution-Free Prediction Intervals Under Covariate Shift, With an Application to Causal Inference.

---

> > > > ### Comment · Reviewer_SpYL · 2025-08-07
> > > >
> > > > Thank you for your clarifications, i have just one last comment.
> > > >
> > > > - When you say that $\mathcal{P}_{cal}$ is the distribution over calibration states, you mean that you consider the empirical distribution of the states from the calibration dataset that you have? If that is the case, I may have some concerns, related to the fact that you write in your Q2 that you have $n_1$ iid calibration states

---

> > > > > ### Author Response · Authors · 2025-08-08
> > > > >
> > > > > We appreciate the reviewer’s feedback. We denote $\mathcal{P}\_{cal}$ as the expectation of the empirical distribution of the data in the original replay buffer which is our calibration dataset. In our first sentence, we assume $n\_1$ i.i.d. calibration states are drawn from $\mathcal{P}\_{cal}$, which can be achieved by randomly sampling from the replay buffer.

---

> ### Comment · Reviewer_SpYL · 2025-08-08
>
> I thank the authors for their response. Just a last note, the samples from the buffer are conditionally iid, and for completeness, it should be sampling with replacement.
>
> I will update my evaluation in the coming days after re-reading the paper and the rebuttal. However, I strongly suggest the authors to incorporate the feedback, provide a proof strategy in the main text, and better clarify some of the points that were discussed. Also better structure the proofs in the appendix would greatly help.
>
> I believe most of the theoretical weaknesses have been addressed, including the comparison with Foffano. However, it is still unclear the scalability to high dimensional problems, and large values of $k$, especially in the off-policy setting. While the authors promised to include high-dimensional settings in one of their comments, it is unclear at the moment if that is feasible. Therefore that remains a weaknesses to me.

---

> ### Author Response · Authors · 2025-08-08
>
> Thank you for your follow-up comments.
> - Regarding your first note, in our paper, the samples from the replay buffer are indeed drawn **with replacement**.
> - We appreciate your suggestions on improving the paper. As promised in our rebuttal, we will incorporate the feedback and improve the structure and clarity of both the main paper and the Supp.
> - Regarding your last point on  the scalability to  large values of $k$, in practice we do not adopt excessively large values of $k$. Please refer to our response to Q3 of Reviewer #1 and **Rebuttal Table B** for details in both the on-policy and **off-policy** settings.
> - Regarding the scalability to high-dimensional problems,  for sure our method is **readily applicable** to this setting, as long as the underlying DRL algorithm can handle high dimensional settings.  This is because our method is  a **CP framework**, and CP can automatically incorporate any statistical and machine learning models or algorithms.
> - In addition, we have been actively conducting experiments in high-dimensional scenarios, with preliminary results reported in **Rebuttal Table D**.
>
> **Rebuttal Table D.** We extend Example 1 to a 50-state setting, where each feature is binary. The action space is \{0,1\}, affecting transitions of only one state. Rewards follow the same distribution as in Example 1. We apply quantile temporal difference (QTD) learning with linear regression and a ridge penalty to mitigate overfitting. The behaviour policy specifies the probabilities of transferring from $x_1=1$ to $x_1=2$ and $x_1=2$ to $x_1=1$ are 0.4 and 0.8, respectively. The target policy is identical to the behavior policy. Experiments are run for $k = 1, \dots, 5$, each repeated 50 times, reporting the mean and standard deviation of coverage probability (cov) and interval length (len).
>
> ||||||||
> |-|-|-|-|-|-|-|
> |cov|k=1|k=2|k=3|k=4|k=5|QTD|
> |ξ=0.7|0.874(0.016)|0.914(0.011)|0.930(0.009)|0.941(0.009)|0.946(0.009)|0.809(0.031)|
> |ξ=0.9|0.866(0.017)|0.905(0.011)|0.922(0.008)|0.932(0.010)|0.940(0.010)|0.809(0.031)|
>
> ||||||||
> |-|-|-|-|-|-|-|
> |len|k=1|k=2|k=3|k=4|k=5|QTD|
> |ξ=0.7|7.394(0.259)|8.145(0.173)|8.480(0.160)|8.711(0.163)|8.878(0.223)|6.770(0.468)|
> |ξ=0.9|7.250(0.272)|7.969(0.194)|8.306(0.155)|8.520(0.179)|8.714(0.217)|6.770(0.468)|

---

### Official Review · Reviewer_dgcF · 2025-06-30

**Clarity:** 3
**Significance:** 3
**Originality:** 4
**Rating:** 5
**Confidence:** 3

**Summary:**

This article considers conformal prediction for policy evaluation in the discounted infinite horizon RL setting. They introduce a new algorithm implementing several trick to address the challenges, namely Pseudo-return construction with Distributional RL to address the unobservable exact returns, Experience replay for the temporal dependence, and Time-Aware Weighted Subsampling for the distribution shift induced. They provide an asymptotic lower bound on coverage as a theoretical guarantee and experiments for empirical validation.

**Questions:**

- Can you detail briefly how this article improves on the current literature ? Mainly:
  - (1) What are the differences in modeling assumption ?
  - (2) How does the scalability compares ?

- Algorithms such as QTD are proven to converge (see [1]), with theoretical guarantees on the limit and the true distribution. Isn’t it enough to get asymptotic prediction intervals ? If it isn’t, could you give a brief explanation on why ? If it is, are they are any other reasons to not use it, other than empirical performances ?

[1] Rowland, Mark, et al. "An analysis of quantile temporal-difference learning." Journal of Machine Learning Research 25.163 (2024): 1-47.

**Ethical Concerns:**

["NO or VERY MINOR ethics concerns only"]

**Final Justification:**

The ideas are original and the contribution is overall significant. Furthermore, the authors addressed all the concerns of the reviewers with their rebuttals.

**Limitations:**

yes

**Paper Formatting Concerns:**

none.

**Quality:**

4

**Strengths And Weaknesses:**

Strengths :
- Combines ideas stemming from different theories to address all the challenges posed by the problem they consider.
- The paper is well written, well referenced and all the different ideas are well introduced in the first sections.
- The experiments are reproducible with the code provided.

Weaknesses :
- It is not entirely clear how this paper positions itself in the literature of Conformal Prediction for RL. The paragraph "Conformal Prediction" in section 1.1 Related Work mentions some other papers but do not explain with them. The sentence "existing methods are limited by scalability issues, strong modeling assumptions, or the curse of the horizon" could be worth given more details.
- In several parts of the paper, such as in Section 1 Introduction line 37, there is a mention of finite-sampling coverage guarantees. Yet to the best of my understanding, this paper only provide asymptotic bounds, as mentioned in the "Contribution" paragraph.
- Minor remarks:
	- In the code provided in the Supplementary Material, some headers are written in Chinese. English would be appreciated for better anonymity and understanding of the code.
	- In the Algorithms, the sets $\mathcal I$ are not explicitly defined.
	- I don’t understand what the paragraph "Off-Policy Setting" in section 3.2 is trying to explain, behind the fact that return distribution can be estimated directly by algorithms such as QTD.

---

> ### Author Rebuttal · Authors · 2025-07-31
>
> # Author response to reviewer dgcF
>
> We sincerely thank Reviewer dgcF for the recognition of our work and for providing constructive comments.
>
> ## Q1: Can you detail briefly how this article improves on the current literature ?
>
> We thank the reviewer for raising this important question regarding the positioning and contribution of our method within the conformal prediction literature for RL. We agree that this is an important point and have revised the Introduction section accordingly.
>
> - **Positioning in Conformal RL Literature.** Our work is the first to tackle **infinite-horizon** off-policy prediction in general reinforcement learning settings for **any state and action spaces** using conformal prediction. Most existing conformal RL methods concentrate on **finite-horizon** scenarios or **finite state or action spaces**, as noted in [1] and [5]. Our approach extends conformal prediction to the infinite-horizon setting while preserving asymptotic coverage guarantees, making it both methodologically novel and technically challenging.
>
> - **Differences in Modeling Assumptions.** As mentioned above, the primary distinction lies in the infinite-horizon framework and the consideration of more general state and action spaces. Furthermore, while most existing methods require access to complete trajectories, our approach remains effective even when only partial trajectory fragments are available, provided that a $k$-step pseudo return can be constructed.
>
> - **Scalability.**   Prior conformal RL methods typically address distribution shifts between behavior and target policies through trajectory-level importance weighting, which exhibits computational inefficiency scaling for  trajectories with increasing horizon. In contrast, our approach utilizes $k$-step pseudo returns and address distribution shift only through a $k$-step importance weighting, therefore it can be implemented efficiently in infinite-horizon settings and continuous tasks, as demonstrated by our experiments on the MountainCar environment.
>
> We will add a more detailed discussion of these points in the revised paper.
>
> ## Q2: Since algorithms like QTD are proven to converge with guarantees on the limiting distribution, isn't that sufficient?
>
> We appreciate this valuable comment. Our response is structured in three parts:
>
> 1. **Limiting distribution.** While QTD is proven to converge to some limiting distribution, this limiting distribution is not guaranteed to be the true return distribution. Please see the last paragraph in Section 4 of [4]. Consequently, this convergence does not provide any guarantee on the asymptotically validity of QTD-based prediction intervals.
>
> 2. **Tabular settings.**  [4] proved the convergence of QTD only in the contex of   finite state and action spaces. To the best of our knowledge, no existing distributional RL algorithm has been proven to consistently recover the true return distribution in general settings with continuous state and action spaces.
>
> 3. **Function approximation.** In continuous state and action spaces, distributional RL methods necessarily employ function approximation for return distribution estimation. The theoretical validity of such approaches is inherently contingent upon the accuracy of the modeling assumptions, rendering them susceptible to potential model misspecification. Conformal prediction helps address this by providing model-agnostic prediction intervals, thereby improving the robustness and reliability of uncertainty quantification in practice.
>
> We will incorporate the above discussions in the ''Limitations of DRL" paragraph.
>
> ## Q3: Clarification on Finite-Sample and Asymptotic Coverage Guarantees
>
> Thank you for pointing this out. We apologize for the confusion. You are absolutely right that our current theoretical results establish asymptotic coverage guarantees. We will revise the wording in lines 37 and 75 in  the Introduction section, and in line 323 in the  Conclusion section to clarify this point.
>
> ## Q4: Minor Remarks
>
> Thank you very much for your careful reading. We will incorporate the following changes in the revised paper:
>
> - We will rewrite the code headers in English to ensure better anonymity and clarity.
>
> - In Algorithm 1, $\mathcal{I}$ refers to the index of the transition $(S_{i,t}, A_{i,t}, R_{i,t}, S_{i,t+1})$ in the training dataset. We will explicitly add this definition in our paper for clarity.
>
> - The QTD algorithm presented in Algorithm 1 of [4] is designed for the **on-policy** setting. Extending QTD to the **off-policy** setting requires careful modifications to account for the distribution shifts between the behavior policy used to collect data and the target policy.
>
>   Specifically, in the on-policy case, QTD estimates the return distribution conditioned on the initial state, $\widehat{\eta}^{\pi}(s)$, using the following iterative update:
>   $$
>   \theta(s,i) \leftarrow \theta(s,i)+\alpha\frac{1}{m}\sum_{j=1}^{m}\left[\tau_i - I(r+\gamma\theta(s^\prime,j)-\theta(s,i)<0) \right],
>   $$
>   where $\theta(s, i)$ denotes the $i$-th quantile of $\widehat{\eta}^{\pi}(s)$ and $(s, a, r, s')$ is sampled under the behavior policy $\pi$, which in the on-policy case coincides with the target policy.
>
>   In contrast, the off-policy setting requires estimating the return distribution conditional on both the state and the action, $\widehat{\eta}^{\pi}(s, a)$, using the modified update:
>   $$
>   \theta(s,a,i) \leftarrow \theta(s,a,i)+\alpha\frac{1}{m}\sum_{j=1}^{m}\left[\tau_i - I(r+\gamma\theta(s^\prime,a^\prime,j)-\theta(s,a,i)<0) \right],
>   $$
>   where $\theta(s, a, i)$ is the $i$-th quantile of $\widehat{\eta}^{\pi}(s, a)$, and $(s, a, r, s')$ is sampled from the behavior policy $\pi_b$. Crucially, $a'$ is drawn from the target policy $\pi$, and the result is marginalized over the action space according to $\pi$. This added complexity arises from the need to correct for the action distribution mismatch between the behavior and target policies.
>
>   This distinction is analogous to the difference
>   between **the TD algorithm** for estimating the state-value function $V^{\pi}(s)$ and **the SARSA algorithm** for estimating the state-action value function $Q^{\pi}(s,a)$. We will clarify this point in the revised paper and refer readers to relevant literature, including [2] and [3] for more details on distributional off-policy evaluation.
>
> # Reference
>
> [1] Conformal off-policy evaluation in markov decision processes.
>
> [2] Distributional Off-policy Evaluation with Bellman Residual Minimization.
>
> [3] Distributional Off-Policy Evaluation in Reinforcement Learning.
>
> [4] An analysis of quantile temporal-difference learning.
>
> [5] Conformal off-policy prediction.

---

> > ### Comment · Reviewer_dgcF · 2025-08-05
> >
> > "I thank the authors for their response, my concerns have been addressed. I will maintain my rating, but I will continue to follow the discussion with reviewer SpYL, who raised several important weaknesses."

---

> > > ### Author Response · Authors · 2025-08-05
> > >
> > > Dear Reviewer dgcF,
> > >
> > > Thank you for your feedback and for acknowledging our response. We sincerely appreciate your decision to maintain the rating and your continued engagement in the discussion.
> > >
> > > Best, Authors

---

### Official Review · Reviewer_g7gT · 2025-07-02

**Clarity:** 3
**Significance:** 3
**Originality:** 3
**Rating:** 5
**Confidence:** 3

**Summary:**

This paper proposes a conformal prediction (CP) framework for constructing distribution-free prediction intervals (PIs) for infinite-horizon on-policy and off-policy evaluation in RL. To do so, the authors propose a i) pseudo-return construction using truncated rollouts and learned return distributions (inspired by k-steps TD learning); ii) experience replay (to decorrelate temporally linked transitions) and weighted subsampling (to correct distribution shifts) to restore approximate exchangeability. Moreover, the authors propose coverage guarantees that depend on the Wasserstein distance between true and estimated return distributions. Eventually, empirical evaluations are provided on synthetic and standard environments (Mountain Car, etc.).

**Questions:**

Could you compare against more methods, particularly from recent conformal off-policy evaluation literature and risk-aware RL ?

Apart from what you wrote line 228 about Zhang et al 2023, why not implementing Foffano et al. (2023?

Can you further elaborate on adaptive or cross-validated strategies for k, or propose heuristics ?

I am very likely to update my score according to other reviews and answers to all feedbacks.

**Ethical Concerns:**

["NO or VERY MINOR ethics concerns only"]

**Final Justification:**

The rebuttal has addressed concerns I had and reading the concerns raised in other reviews, I have decided to keep my score, which still leans towards an acceptance.

**Limitations:**

The authors did not discuss broader impacts of the paper and their answer to that in the checklist (10. Broader impacts) is rather odd: “The manuscript does not contain a Broader-Impact discussion of how autonomous tool-using LLMs might be beneficial or misused.” Is that a copy paste from another NeurIPS submission?

**Quality:**

3

**Strengths And Weaknesses:**

1) Strengths

The research problem is very relevant to the NeurIPS community. Uncertainty quantification in RL is important and few existing methods offer finite sample guarantees. The authors propose an integration of conformal prediction with distributional RL with a plug-and-play method of any return distribution estimator.

The coverage bounds are derived under approximate exchangeability using Wasserstein distances. These bounds bring nuanced insights of model and weight estimation errors. The difference in the bounds of Theorem 1 and 2 is well explained and the intuition on differences between on and off-policy is well explained. Moreover, the empirical results across on policy and off-policy scenarios show clear improvements in coverage over standard DRL-QR, which are quantile methods.

The paper is clearly written with very well articulated sections, well defined related work and problem setting. I did not check the novelty w.r.t related work however.

2) Weaknesses

Empirical evaluation: A few things that to my mind represent weaknesses in the experiment section. The choice of the step size k in pseudo-returns is a little underexplored (experiments run with k=1...5 only). The authors mention aggregating different PIs over k in the conclusion but don’t offer a principled selection method or guidelines. Also, the only comparisons are to DRL-QR and KDE-QR. There is no comparison to recent methods in conformal RL like Foffano et al. (2023), Zhang et al. (2023). Moreover, the settings considered in the experiments are mostly low-dimensional.

---

> ### Author Rebuttal · Authors · 2025-07-31
>
> # Author Response to Reviewer g7gT
> We sincerely thank Reviewer g7gT for the recognition of our work and for providing constructive comments.
>
> ## Q1: Add comparison with Foffano et al. (2023), Zhang et al. (2023) and risk-aware RL.
>
> We appreciate the constructive feedback provided. Following the suggestion, we add a comparison with Foffano et al. (2023) in **Rebuttal Table A**, where our method demonstrates superior performance in terms of both coverage probability and average length.
>
> We would like to highlight that Foffano et al. (2023) focused on **finite-horizon** Markov Decision Processes (MDPs) and addressed the distributional shift problem
> using estimates of  importance sampling weights. The importance sampling weight function (see Equation (12) of Foffano et al. (2023)) is
> $$
>  w(x,y) = E_{\tau\sim P_{\tau|X=x,Y=y}^{\pi_b}} \left[ \frac{\prod_{t=1}^H\pi(a_t|x_t)}{\prod_{t=1}^H\pi^b(a_t|x_t)} \right],
> $$
> where $H$ is the time horizon,  $\tau =  (x_1, a_1, r_1, x_2, a_2, r_2, \ldots, x_H, a_H, r_H) $ is a  trajectory of length $H$,  $x$ is the initial state, $y$ denotes the $H$-step cumulative return under the behavior policy $\pi^b$, and $\pi$ is the target policy.
> Foffano et al. (2023)  repeatedly emphasized the difficulty caused by the **curse of horizon**  in the concluding paragraphs of their Sections V-B.1, V-B.2, and VI-D.2. In contrast, our approach is specifically designed for the **infinite-horizon** setting, where we construct pseudo-returns based on truncated rollouts. This leads to a fundamentally different framework that avoids the compounding challenges associated with long-horizon importance weighting.
>
>    **Rebuttal Table A.** We compare the performance on Example 1 with a fixed horizon of 20. For Foffano’s method, we follow their gradient-based approach to train the likelihood ratio model $w(x, y)$ via linear regression, and apply WCP to construct prediction intervals. For our method, we replace the nonconformity score with the double-quantile score from Foffano et al.(2023), with $\xi$ set to 0.5 and 0.6. Each experiment is repeated 100 times, and we report the mean and standard deviation of coverage probability (cov) and interval length (len) when the nominal coverage level is  90%.
>
> ||||
> |-|-|-|
> ||cov|len|
> |Foffano|0.85(0.01)|6.51(0.11)|
> |our proposal|||
> |k=2,$\xi$=0.5|0.91(0.01)|7.67(0.15)|
> |k=3,$\xi$=0.5|0.91(0.01)|7.77(0.13)|
> |k=2,$\xi$=0.6|0.92(0.01)|8.00(0.16)|
> |k=3,$\xi$=0.6|0.92(0.01)|8.02(0.15)|
>
> As for Zhang et al. (2023), their COPP algorithm is specifically designed for contextual bandits and can only be extended to **short-horizon** decision-making scenarios (e.g., $H=5$). In such short-horizon settings, employing truncated rollouts, as done in our method for infinite-horizon problems, is generally unnecessary. Moreover, the resampling-and-matching strategy proposed in COPP is limited to finite discrete action spaces, whereas our method is applicable to **both discrete and continuous** action spaces. Therefore, we consider the two methods to be fundamentally different in scope: COPP targets short-horizon, discrete-action problems, while our approach is designed for more general and challenging infinite-horizon settings. As such, a direct comparison is not applicable.
>
> Finally, we thank the reviewer for pointing us to the literature on **risk-aware RL**. We acknowledge that risk-aware RL constitutes an important line of work aiming to optimize risk-sensitive objectives, such as CVaR or variance-penalized returns (e.g., [1,2]). These methods focus on modeling risk preferences rather than constructing valid uncertainty quantification for prediction intervals. We will incorporate a discussion on this literature in the revised ``Related Work'' section.
>
> ## Q2: Apart from line 228 about Zhang et al 2023, why not Foffano et al. (2023)?
>
> We appreciate the reviewer's insightful question, which prompted deeper reflection on the relationship between our method and that of Foffano et al. (2023). According to our understanding, the method proposed by Foffano et al. (2023) consists of two key components:
>
> - A new construction of nonconformity scores, such as the double-quantile score;
> - The estimation of importance sampling weights, as described in our response to Q1.
>
> These components are combined within the standard Weighted Conformal Prediction (WCP) framework. As such, implementing  Foffano et al. (2023)'s method in our setting on line 228 essentially amounts to implementing WCP. In response, we will add this point in line 228. While this is feasible, we discuss the limitations of WCP in lines 224--230 and provide a rationale for adopting the Weighted Subsampling approach instead, as detailed in lines 215--218.
>
> Moreover, we find their construction of nonconformity scores particularly insightful. Given that our framework permits flexible selection of nonconformity scores (Step 3), their double-quantile score can be seamlessly integrated. We have presented supporting simulations in **Rebuttal Table A**. In response, we will add this point in line 153.
>
>
> ## Q3: Add exploration of $k$, adaptively choosing $k$, and aggregation of PIs across $k$
>
> ### Q3-1: Add exploration of the step size $k$
>
> Following your suggestion,  we conducted additional  experiments with $k = 6, 7, 8$ in Example 1. The results presented in **Rebuttal Table B** demonstrate that larger $k$ values consistently lead to overcoverage and consequently wider prediction intervals of our method. This empirical finding is consistent with our theoretical analysis in Section 4 (Theorems 1 and 2), which establishes that the coverage probability incorporates an additional slack  or an error term $\Lambda(\widehat{w},\widehat\eta^{\pi})$. The error component originates from two distinct sources:
> (i) the estimation error of the weight function $\widehat{w}$, and
> (ii) the approximation error of the return distribution $\widehat\eta^{\pi}$.
>
> This creates a **trade-off**: a larger $k$ reduces the approximation error in estimating $\widehat\eta^{\pi}$ but increases the difficulty of accurately estimating the off-policy weights, particularly under significant distributional shift. Based on our empirical findings, we recommend using $k = 2$ or $3$ to balance these competing factors. We will add the above discussion on the role of $k$ in the Theorem section.
>
> **Rebuttal Table B.** We conduct the experiments for Example 1 across $k=1,2,\ldots,8$. Each experiment is repeated 100 times, and we report the mean and standard deviation of coverage probability (cov) and interval length (len) when the nominal coverage level is  90%.
>
> ||||||||||
> |-|-|-|-|-|-|-|-|-|
> |**on policy**|||||||||
> |cov|k=1|k=2|k=3|k=4|k=5|k=6|k=7|k=8|
> |$\xi$=0.8|0.87(0.01)|0.90(0.01)|0.91(0.01)|0.92(0.01)|0.92(0.01)|0.93(0.01)|0.94(0.01)|0.94(0.01)|
> |len|k=1|k=2|k=3| k=4|k=5|k=6|k=7|k=8||
> |$\xi$=0.8|7.78(0.10)|8.24(0.10)|8.56(0.13)|8.78(0.14)|9.00(0.15)|9.15(0.19)|9.31(0.23)|9.50(0.22)|
> |**off policy**|||||||||
> |cov|k=1|k=2|k=3|k=4|k=5|k=6|k=7|k=8|
> |$\xi$=0.8|0.87(0.01)|0.91(0.01)|0.92(0.01)|0.92(0.01)|0.93(0.01)|0.93(0.01)|0.93(0.01)|0.94(0.02)|
> |len|k=1|k=2|k=3| k=4|k=5|k=6|k=7|k=8|
> |$\xi$=0.8 |7.57(0.10)|8.13(0.11)|8.47(0.14)|8.67(0.14)|8.90(0.17)|9.02(0.18)|9.20(0.18)|9.26(0.20)|
>
> ### Q3-2: Adaptive or cross-validated strategies for choosing $k$
>
> We appreciate the reviewer’s suggestion on exploring adaptive or cross-validated strategies for selecting $k$. While these approaches are conceptually straightforward in on-policy settings where target policy validation data are readily available, they present significant challenges in off-policy settings due to the absence of independent validation data sampled from the target policy. This limitation makes performance-based tuning of $k$ difficult under distribution shift.
>
> ### Q3-3: Aggregating PIs across $k$
>
> We also thank the reviewer for raising the interesting question of aggregating prediction intervals across different $k$ values. Since these intervals are correlated, aggregation is nontrivial. A promising direction is to construct a unified prediction region by combining the corresponding p-values, leveraging the connection between prediction intervals and hypothesis testing. Methods such as the **Cauchy Combination Test** (e.g., [4,5,6]), which are robust to arbitrary dependencies, offer a viable approach. We will include this discussion in the Conclusion section.
>
> ## Q4: The settings considered in the experiments are mostly low-dimensional.
>
> Our experiments primarily focus on low-dimensional settings to clearly demonstrate the method’s performance. Due to time constraints during the rebuttal period, we will incorporate high-dimensional experiments in the revised version.
>
> # References
>
> [1] Risk-sensitive and robust decision-making: a cvar optimization approach.
>
> [2] Risk-constrained reinforcement learning with percentile risk criteria.
>
> [3] Conformal off-policy evaluation in markov decision processes.
>
> [4] ACAT: a fast and powerful p value combination method for rare-variant analysis in sequencing studies.
>
> [5] Cauchy combination test: a powerful test with analytic p-value calculation under arbitrary dependency structures.
>
> [6] Multi‐split conformal prediction via Cauchy aggregation.
>
> [7] Conformal off-policy prediction.

---

> > ### Comment · Reviewer_g7gT · 2025-08-05
> > **thank you**
> >
> > Thank you for your additional experiments and clarifications you brought. The rebuttal has addressed concerns I have, and I am happy to keep my score.

---

> > > ### Author Response · Authors · 2025-08-06
> > >
> > > Dear Reviewer g7gT,
> > >
> > > Thank you for your feedback. We’re glad to know that our responses have addressed your concerns and that you’ve decided to maintain your score. We sincerely appreciate your time and consideration.
> > >
> > > Best, Authors

---

### Decision · Program_Chairs · 2025-09-17

**Decision:**

Accept (poster)

**Comment:**

The paper proposes a conformal prediction framework for off-policy evaluation. All reviewers feel positive about the paper, as it studies an important topic that the NeurIPS community cares about and takes an important initial step. Reading through the paper and the reviews, the AC has some comments and suggestions about the framing of the work:

1. Standard OPE formulation often cares about the expected return, which is described as "insufficient" in intro. However, CP guarantees are not strictly stronger than mean estimation guarantees, as the latter cannot be induced from the former even with infinite amount of data.

2. The authors distinguish the submission from prior works in paper and rebuttal by (1) the infinite-horizon settings (vs. finite-horizon), and (2) accounting for misspecification (and prior works somehow don't do as well). For (1), it is mentioned that Foffano et al suffer the curse of horizon which is avoided by the current work. The AC's feeling is that you can't have both simultaneously, and the paper does not really solve these issues in some fundamental sense, but rather provide some soft mitigations using standard ideas from the RL toolbox (such as return truncation and multi-step learning). In particular, Theorems 1 and 2 has the error of tail distribution $\eta(s)$ (which I assume partially stems from model misspecification) discounted by $\gamma^k$. To say you do much better than prior works on model misspecification you'd want large $k$ so that this term is negligible, but any non-constant $k$ would cause curse of horizon in the off-policy setting, making the approach not fundamentally different from the situation in Foffano et al's work.